# Manipulation of nonlinear optical responses in layered ferroelectric niobium oxide dihalides

Liangting Ye[1,9], Wenju Zhou[2,9], Dajian Huang[2], Xiao Jiang[1], Qiangbing Guo[3], Xinyu Cao[4], Shaohua Yan[5,6], Xinyu Wang[2], Donghan Jia[2], Dequan Jiang[2], Yonggang Wang[2], Xiaoqiang Wu[7], Xiao Zhang[3], Yang Li[1]✉, Hechang Lei[4,5], Huiyang Gou[2]✉ & Bing Huang[1,8]✉

Realization of highly tunable second-order nonlinear optical responses, e.g., second-harmonic generation and bulk photovoltaic effect, is critical for developing modern optical and optoelectronic devices. Recently, the van der Waals niobium oxide dihalides are discovered to exhibit unusually large second-harmonic generation. However, the physical origin and possible tunability of nonlinear optical responses in these materials remain to be unclear. In this article, we reveal that the large second-harmonic generation in $NbOX_2$ ($X$ = Cl, Br, and I) may be partially contributed by the large band nesting effect in different Brillouin zone. Interestingly, the $NbOCl_2$ can exhibit dramatically different strain-dependent bulk photovoltaic effect under different polarized light, originating from the light-polarization-dependent orbital transitions. Importantly, we achieve a reversible ferroelectric-to-antiferroelectric phase transition in $NbOCl_2$ and a reversible ferroelectric-to-paraelectric phase transition in $NbOI_2$ under a certain region of external pressure, accompanied by the greatly tunable nonlinear optical responses but with different microscopic mechanisms. Our study establishes the interesting external-field tunability of $NbOX_2$ for nonlinear optical device applications.

The responses of materials under the optical field are the basis of many important physical phenomena and detection techniques. Among numerous materials, compounds without inversion symmetry can exhibit various second-order nonlinear optical (NLO) responses. The second-harmonic generation (SHG), referred the effect that the frequency of light is doubled when it passes through a material, is a typical example of the NLO effect, which has been not only used to characterize the basic properties of materials, such as structural symmetry[1,2], magnetic ordering[3,4], and polarized domains[5], but also used to design electro-optical devices, such as frequency conversions[6,7], electro-optic modulators[8,9] and optical switches[10]. Besides the SHG, the shift current, referred the direct current induced by the change of wave functions when the electrons are excited from the valence band to the conduction band, is another important NLO

[1]Beijing Computational Science Research Center, Beijing 100193, China. [2]Center for High Pressure Science and Technology Advanced Research, Beijing 100193, China. [3]Department of Electrical and Computer Engineering, National University of Singapore, Singapore, Singapore. [4]State Key Laboratory of Information Photonics and Optical Communications & School of Science, Beijing University of Posts and Telecommunications, Beijing 100876, China. [5]Department of Physics and Beijing Key Laboratory of Optoelectronic Functional Materials & MicroNano Devices, Renmin University of China, Beijing 100872, China. [6]Key Laboratory of Quantum State Construction and Manipulation (Ministry of Education), Renmin University of China, Beijing 100872, China. [7]School of Mechanical Engineering, Chengdu University, Chengdu 610106, China. [8]Department of Physics, Beijing Normal University, Beijing 100875, China. [9]These authors contributed equally: Liangting Ye, Wenju Zhou. ✉e-mail: liyang2020@csrc.ac.cn; huiyang.gou@hpstar.ac.cn; Bing.Huang@csrc.ac.cn

phenomenon. The shift current is the main mechanism of the bulk photovoltaic effect (BPVE)[11,12], which is promising to overcome the Shockley–Queisser limit of solar energy conversion[13].

Searching materials with large and greatly tunable NLO responses is what we have long sought for various NLO device applications[8]. The advent of low-dimensional materials provides another opportunities to find candidates with large second-order NLO responses, such as transition-metal-dichalcogenide (TMD) layers[2,14] and nanotubes[15], twisted boron nitrides[16], and 2D multiferroics[17], which is more suitable for the design of nanoscale electronic and photonic devices compared with traditional bulk materials. Very recently, the layered ferroelectric (FE) material niobium oxide dihalides $NbOX_2$ ($X$ = Cl, Br, and I) have attracted plenty of interests due to their large, anisotropic, and even layer-independent SHG responses[18,19], significantly different from many known 2D NLO materials[2,20,21]. Consequently, the $NbOCl_2$ thin-film can be used to design a spontaneous parametric down-conversion quantum light source with a record performance[19]. Interestingly, although holding different halides, the observed SHG strength in $NbOCl_2$ and $NbOI_2$ are quite similar[18,19]. Given the rapid progress of exciting experiments, the microscopic origin of SHG effects in $NbOX_2$ remains to be largely known, e.g., it is unclear whether the NLO effects of $NbOX_2$ are strongly coupled with their ferroelectricity. On the other hand, developing effective means to manipulate the NLO responses is highly desired for FE $NbOX_2$, which is the key step for realizing $NbOX_2$-related optical sensing, optical computing, and optical switch.

In this work, we reveal the physical origin of large NLO responses observed in $NbOX_2$ using density functional theory (DFT) based on NLO calculations (see "Method"). We discover that the large SHG observed in $NbOCl_2$ is mainly contributed by the double-photon resonances between occupied anion $p$ orbitals and empty Nb $4d$ orbitals, partially originating from band-nesting-induced large joint density of states (JDOS). Besides SHG, the $NbOCl_2$ can exhibit very different strain-dependent shift current responses under different polarized light, originating from the selective optical transitions between different orbitals that hold anisotropic deformation

potentials. Combining with the single-crystal X-ray diffraction and SHG measurements, we observe that $NbOCl_2$ can experience an unusual structural phase transition from the FE phase to the antiferroelectric (AFE) phase while $NbOI_2$ undergoes a phase transition from the FE phase to the paraelectric (PE) phase in a certain region of external pressure, accompanied by greatly tunable material polarization and NLO effects. Finally, we demonstrate that the SHG (shift current) response in $NbOX_2$ is insensitive (sensitive) to the different $X$.

## Results

### Structural and electronic properties of $NbOX_2$

Niobium oxide dihalides $NbOX_2$ are a series of the van der Waals (vdW) layered materials[22–24]. As shown in Fig. 1a, the individual vdW layers are ABC stacked along $a$ direction, crystallized to a monoclinic structure with the $C2$ space group. Within each vdW layer, Nb atoms are inside the edge sharing distorted octahedra chains which mutually connect via O atoms along $b$ direction. Compared with other transition-metal oxide dihalides[25,26], the combined displacements of Nb atoms along $c$, $b$ directions create a unique crystal structure of $NbOX_2$: (i) the Nb atoms are dimerized along $c$ direction (i.e., $d_l > d_s$), resulting in a noticeable Peierls distortion, accompanied by a metal-to-insulator transition (Supplementary Fig. 1); (ii) the off-center displacements of Nb atoms along $b$ direction causes the separation of positive and negative charge centers, further forming spontaneous polarization ($P$) and breaking the inversion symmetry (Fig. 1b, *left* panel). Interestingly, whether the appearance of FE phase could enhance the NLO responses is still an open question[27,28], which may be material dependent. In principle, the shift of several adjacent Nb atoms to the opposite directions or vanishing polarization may create other metastable phases, i.e., AFE phase or PE phase (Fig. 1b, *right* panel), although the possible metastable phases have never been observed in the experiments. In contrast to the FE phase, the inversion symmetry of high symmetric $NbOX_2$ may forbid all the second-order NLO responses.

For $NbOX_2$ family, the weak interlayer coupling is a unique property that is different from many other 2D materials[19], reflected by the

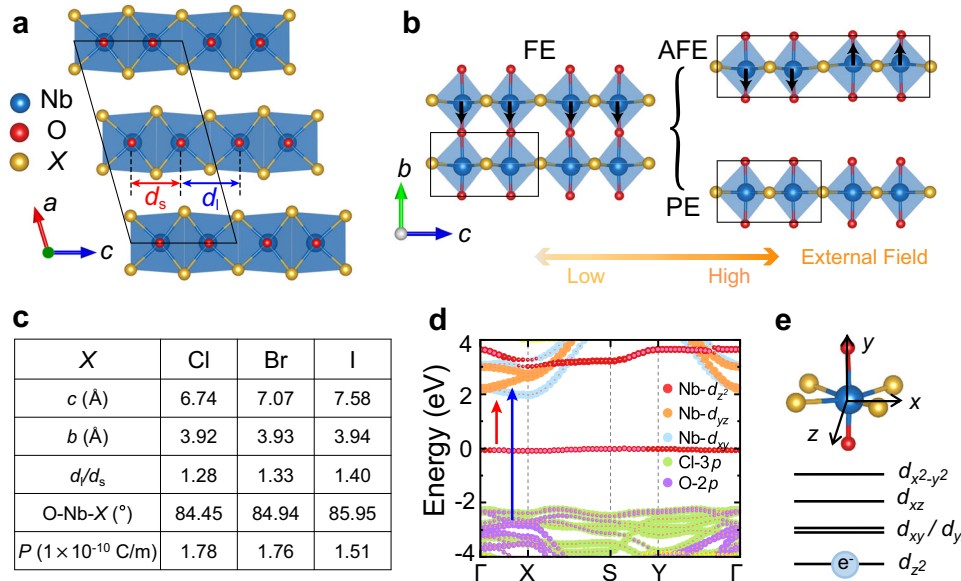

**Fig. 1 | Crystal and electronic properties of $NbOX_2$. a** Front view of the crystal structure of $NbOX_2$. Here, the blue, red, and yellow balls represent Nb, O, and $X$ atoms, respectively. $d_l$ and $d_s$ are the distances between the Nb atomic pairs. black lines indicate the unit cell. **b** Bottom view of the monolayer structure for $NbOX_2$ in FE phase (*left* panel), which may be converted to a possible AFE or PE phase (*right* panel) via applying external approaches, like pressure. **c** Summarization of the calculated parameters of monolayer $NbOX_2$ in the ground-state FE phase. Here,

$c$ and $b$ represent lattice constants, $P$ indicates polarization, O-Nb-$X$ means bond angle, $d_l$ and $d_s$ are distances shown in (**a**). **d** HSE-calculated band structure of monolayer $NbOCl_2$ with orbital projection. Here, the red and blue arrows represent two important optical transitions between Nb $d$–Nb $d$ orbitals and between Nb $d$–Cl $p$ orbitals, respectively. **e** Local geometries of $NbOCl_2$ (*upper* panel) and the schematic energy splitting of Nb-$d$ orbitals (*bottom* panel). Here, the blue ball means an unpaired electron.

| $X$ | Cl | Br | I |
|---|---|---|---|
| $c$ (Å) | 6.74 | 7.07 | 7.58 |
| $b$ (Å) | 3.92 | 3.93 | 3.94 |
| $d_l/d_s$ | 1.28 | 1.33 | 1.40 |
| O-Nb-$X$ (°) | 84.45 | 84.94 | 85.95 |
| $P$ ($1 \times 10^{-10}$ C/m) | 1.78 | 1.76 | 1.51 |

layer-insensitive electronic band structures (Supplementary Fig. 2). Taking monolayer NbOCl₂ as an example, Fig. 1c lists the calculated basic parameters in the ground-state FE phase. As one can see, the lattice constant $c$ increases obviously with the change of $X$ from Cl to I, expanding the space of O₂X₄ octahedra and enhancing the Peierls distortion (i.e., increase of $d_l/d_s$). In contrast, the point-sharing connection of the O₂X₄ octahedra causes the lattice constant $b$ insensitive to $X$. Moreover, the increase of atomic radius with the change of $X$ from Cl to I can also increase O-Nb-$X$ bond angle, weakening the electrical dipole moment for O₂X₄ octahedra, which is consistent with the decreased polarization $P$. The rather different changes of the lattice constants $c$ and $b$ with varying halogen clearly suggest the remarkable crystal anisotropy of NbO$X$₂. In the following, we mainly focus on the NbOCl₂, and the results of other isostructural NbO$X$₂ will be discussed at the end of this paper.

Figure 1d shows the HSE-calculated band structure of monolayer NbOCl₂, which exhibits a feature of indirect bandgap ~1.98 eV[19]. According to the crystal field theory and Jahn–Teller distortion[29], the Nb $d$ orbitals will split into three nondegenerate subgroups (i.e., $d_{z^2}$, $d_{xz}$, $d_{x^2-y^2}$) and a double-degenerate subgroup (i.e., $d_{xy}$ and $d_{yz}$). As the Nb atom has the valence electronic configuration of $4d^4 5s^1$, there is only one unpaired $4d$ electron for each cation Nb⁺⁴ in NbOCl₂, half filling the Nb $4d_{z^2}$ orbitals (Fig. 1e). The dimerization of Nb atoms along $c$ direction, in the form of Peierls distortion, further splits the Nb $4d_{z^2}$ orbitals in energy. This Peierls distortion can create an isolated flat bonding-like Nb $4d_{z^2}$ band around the Fermi level, forming the top of the valence band (VB) with a large density of states in NbOCl₂. Meanwhile, the empty antibonding-like Nb $4d_{z^2}$ orbitals are pushed to ~1 eV above the bottom of the conduction band (CB). Noticeably, the low-energy O $2p$ and Cl $3p$ orbitals mainly contribute to the VB around 2 eV below the Fermi level, well separating from the flat band.

## Unusual SHG responses in NbOCl₂

Figure 2a shows the typical SHG process, where the frequency $\omega$ of light is doubled when it passes through a NLO material. This SHG effect can be described as[30]:

$$P^a(2\omega) = \epsilon_0 \chi^{(2)}_{abc} E^b(\omega) E^c(\omega), \tag{1}$$

where $\boldsymbol{P}(2\omega)$ represents the polarization, $\chi^{(2)}$ is known as SHG susceptibility, $\boldsymbol{E}(\omega)$ indicates the electric field of the incident light, $\epsilon_0$ is the dielectric constant of vacuum, and $a,b,c$ are the indices in Cartesian coordinates. For the non-centrosymmetric materials, the number of independent elements of the three-order tensor $\chi^{(2)}$ is governed by the direct product $\Gamma_P \otimes \Gamma_{EE}$[31]. Due to the symmetry limitation for the second-order susceptibility tensor and the weak interlayer coupling, the non-zero tensor components we are interested in the bulk NbO$X$₂ are almost same in the monolayer NbO$X$₂ (Supplementary Fig. 3 and Supplementary Note 1). For monolayer NbOCl₂ with the $C_{2v}$ point group, the representations $\Gamma_P$ and $\Gamma_E$ can be divided into three irreducible representations: $\Gamma_P = \Gamma_E = A_1 + B_1 + B_2$. For linear susceptibility, we have $\Gamma_P \otimes \Gamma_E = 3A_1 + 2A_2 + 2B_1 + 2B_2$, leading to three independent nonzero components. The direct product $\Gamma_P \otimes \Gamma_E$ can be further divided into a symmetric part $\Gamma^s = 3A_1 + A_2 + B_1 + B_2$ and an anti-symmetric part $\Gamma^a = A_2 + B_1 + B_2$[31]. Hence for the symmetric third-rank tensor, we can write $\Gamma_P \otimes \Gamma^s = 5A_1 + 3A_2 + 5B_1 + 5B_2$, suggesting only five independent nonzero elements in the $\chi^{(2)}$ tensor for monolayer NbOCl₂.

Indeed, the calculated SHG susceptibilities have five nonzero components, i.e., $\chi^{(2)}_{xxy}$, $\chi^{(2)}_{yxx}$, $\chi^{(2)}_{yyy}$, $\chi^{(2)}_{yzz}$, and $\chi^{(2)}_{zyz}$, as shown in Fig. 2b. Among them, the components along the non-polarized directions, i.e., $\chi^{(2)}_{xxy}$ and $\chi^{(2)}_{zyz}$, are small in a large photon energy range. On the contrary, the components along the polarized directions display noticeable values, a feature clearly demonstrating the unique role of ferroelectricity in generating anisotropic SHG responses in NbOCl₂. The component $\chi^{(2)}_{yxx}$ shows great values within the range of 2~4 eV, with a maximum of ~120 pm/V at $\hbar\omega = 2.2$ eV. The component $\chi^{(2)}_{yzz}$ has the similar maximum of ~110 pm/V at the higher position of $\hbar\omega = 4.8$ eV. The most important component is $\chi^{(2)}_{yyy}$, which corresponds to the SHG response induced by the $y$-polarized incident light. Interestingly, the $\chi^{(2)}_{yyy}$ displays the large value over a wide energy region and reaches its

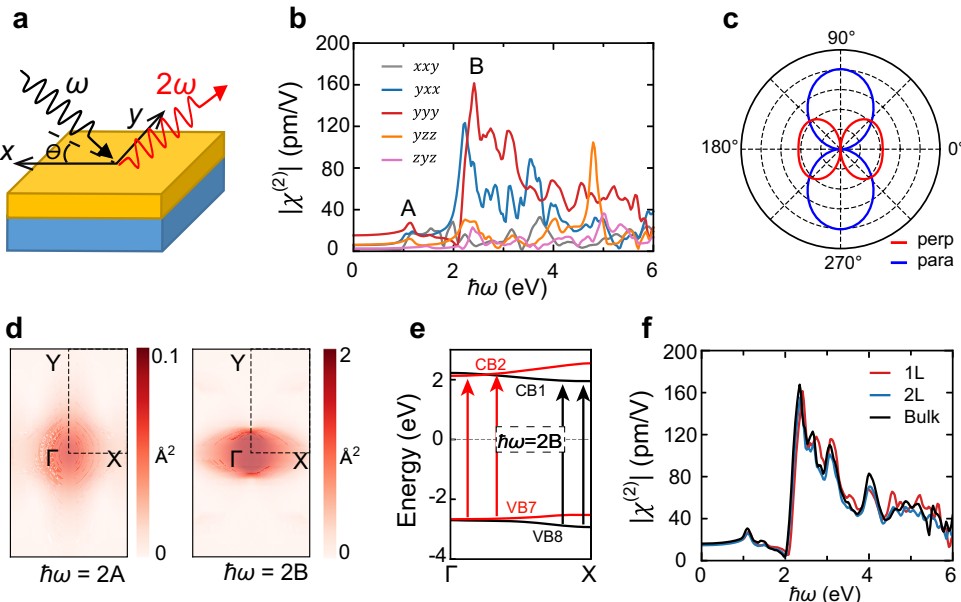

**Fig. 2 | SHG responses in NbOCl₂. a** Schematic diagram of the SHG process. **b** Calculated nonlinear susceptibility $\left|\chi^{(2)}_{abc}\right|$ as a function of incident photon energy. **c** Angle-resolved SHG polarization. Here, the photon energy of the incident light is set to be 2.4 eV, and the red (blue) curve represents the response with the direction perpendicular (parallel) to the polarization of incident light. **d** $k$-resolved optical absorption strength near SHG peaks, representing $d$–$d$ transition (*left* panel) and $p$–$d$ transition (*right* panel) over the first Brillouin zone. **e** Schematic diagram of the strong optical transitions caused by the band nesting along Γ−X line. Here, VB$_n$ (CB$_n$) labels the $n$th valance (conduction) band starting from the top valence (bottom conduction) band. **f** SHG response for monolayer, bilayer, and bulk NbOCl₂.

maximum of ~160 pm/V at $\hbar\omega = 2.4$ eV, denoted as peak B. The strength of peak B is much larger than the peaks of $|\chi^{(2)}|$ in BN sheet (~20 pm/V)[2] and LiNbO$_3$ (~50 pm/V)[32]. Besides peak B, there is also a small peak that appeared at $\hbar\omega = 1.14$ eV in $\chi^{(2)}_{yyy}$, denoted as peak A. To analyze the SHG effect of NbOCl$_2$, we also calculate the angle-resolved SHG polarization of monolayer NbOCl$_2$ under $\hbar\omega = 2.4$ eV, as shown in Fig. 2c. With the azimuth angle ($\theta$) [marked in Fig. 2a] changes, the SHG response exhibits a significant anisotropy and shows a two-fold rotational symmetry, consistent with the symmetry of crystal structure.

We focus on the understanding of the origin of peaks A and B in $\chi^{(2)}_{yyy}$, and similar peaks exist in other nonzero $\chi^{(2)}$ components. In general, the SHG is closely related to the transition dipole moment $r^a_{nm}$ and JDOS[33,34] (see Method). Interestingly, for the $r^a_{nm}$, the calculated results under $y$-polarized incident light exhibit different origins for peak A and peak B. As shown in Fig. 1d (red arrow), the peak A is contributed by the double-photon resonances between the flat VB dominated by the Nb $d_{z^2}$ orbital and bottom of CB dominated by the Nb $d_{xy}$ orbitals, i.e., the $d \rightarrow d$ transition, while the peak B is mainly contributed by the double-photon resonances between the VB below ~−2 eV dominated by Cl 3$p$ and O 2$p$ orbitals and bottom of CB, i.e., the $p \rightarrow d$ transition (blue arrow in Fig. 1d). Figure 2d shows the **k**-resolved optical absorption strength, i.e., $r^y_{nm}r^y_{mn}$, which can characterize the strength of interaction between the system and the electromagnetic wave quantitatively. The absorption strength of peak B is obviously stronger than peak A, which may induce stronger SHG response. Meanwhile, as shown in Fig. 2e, the double-photon resonances dominated peak B is mainly caused by the transitions between the 7th (8th) VB and the 2nd (1st) CB. These bands for optical transitions are almost parallel to each other along the $\Gamma − X$ path, resulting in a significant band nesting effect and creating the singularities in the JDOS. Therefore, it is expected that the singularities in JDOS may play an important role in further enhancing the SHG response at peak B[35].

Furthermore, we have calculated the $\chi^{(2)}_{yyy}$ spectra of NbOCl$_2$ with different layer thicknesses. Different from other vdW layered NLO compounds such as TMDs[2,36] and SnS (Se)[37], where the SHG strength will gradually decrease with increasing the sample thickness. In addition, we have compared the available experimental SHG data with our calculated one. Importantly, our calculated spectrum has a similar curvature to the experimental one[19] in spite of the small difference in the absolute values of SHG caused by the approximation used in

theoretical calculations[35,38]. Therefore, the peak B may account for the observed large SHG in NbOCl$_2$ in experiment. In addition, the exclusion of exciton effect[39] may also result in avoidable differences between our calculations and experimental data, which deserves future investigation.

## Shift current in NbOCl$_2$

Besides the large SHG response, the non-centrosymmetric structure also allows NbOCl$_2$ to generate nonzero shift current density **J** under external electric fields[30]:

$$J^a_{sc} = 2\sigma^{abc}(0;\omega,-\omega)E^b(\omega)E^c(-\omega), \qquad (2)$$

where $\sigma$ is the shift current susceptibility tensor. Similar to SHG, there are only five independent susceptibility components in monolayer NbOCl$_2$ owing to the restriction of crystal symmetry (Supplementary Fig. 4). Figure 3a shows two in-plane $\sigma$ with the most remarkable values. As one can see, the component $\sigma^{yxx}$ caused by the $x$-polarized incident light reaches the maximum when $\hbar\omega \approx 3.5$ eV and reverses its sign when $\hbar\omega > 4.0$ eV. However, the component $\sigma^{yyy}$ induced by the $y$-polarized incident light has large values over a relatively wide range of $4.3 < \hbar\omega < 5.0$ eV.

The $\sigma$ can be considered as an integration of shift vector weighted by optical transition rate over the entire Brillouin zone[27,40]. To illustrate the mechanism of $\sigma$, we have calculated the integrated shift vector $e\bar{R}^{ab}$ and the linear optical absorption spectrum $\varepsilon^{ab}_2$ (see "Method"). Because the phase of transition dipole and Berry connection appeared in shift vector contain the information about the change of the charge center in real space during excitation, the $e\bar{R}^{ab}$ usually can determine the main features of $\sigma$. Indeed, as shown in Figs. 3a and 3b, the spectrum of $e\bar{R}^{yy}$ has a similar shape to that of $\sigma^{yyy}$, and the amplitudes of $e\bar{R}^{yy}$ with positive and negative values are also consistent with that of $\sigma^{yyy}$. On the other hand, the $\sigma^{yxx}$ seems to be mainly determined by the $\varepsilon^{xx}_2$. Especially, the large peak in $\varepsilon^{xx}_2$ is consistent with the peak A of $\sigma^{yxx}$, as shown in Figs. 3a and 3c.

Furthermore, we evaluate the change of $\sigma$ under in-plane strain, and the corresponding results are shown in Fig. 3d, e. Due to the significant anisotropy of NbOCl$_2$, the major source of optical transitions is dependent on the polarization of the incident light. For the $x$-polarized light, the optical transitions are dominated by the coupling between two out-of-plane Nb $d_{z^2}$ states, where the dispersion and

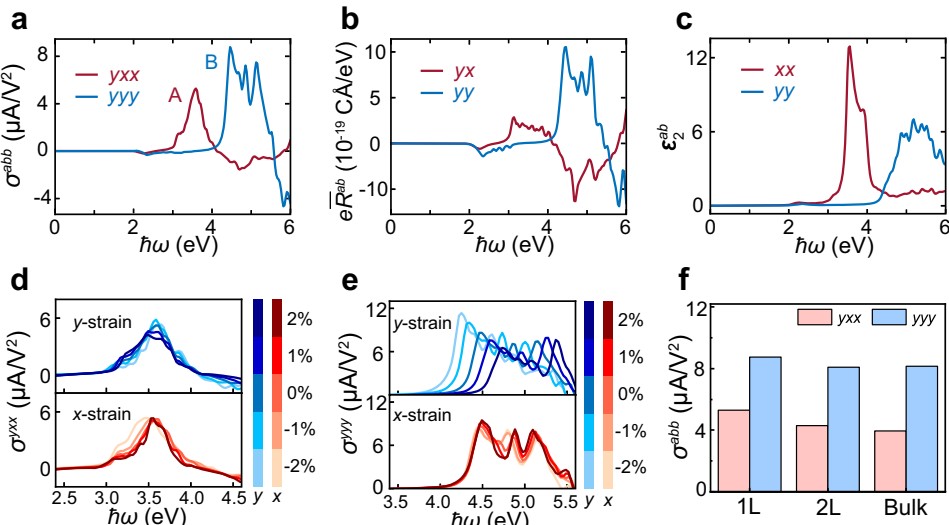

**Fig. 3 | Shift Current in NbOCl$_2$. a–c** Frequency-dependent shift current susceptibility tensor $\sigma^{abb}$, integrated shift vector $e\bar{R}^{ab}$, and optical absorption spectrum $\varepsilon^{ab}_2$ induced by the $y$-polarized and $x$-polarized incident light, respectively. **d, e** Frequency-dependent $\sigma^{yxx}$ and $\sigma^{yyy}$ under uniaxial in-plane strains along the $x$ or $y$ direction, respectively. **f** Comparison of the peak A in $\sigma^{yxx}$ and peak B in $\sigma^{yyy}$ with different layer thicknesses in NbOCl$_2$.

energy gap between these two states are less sensitive to the in-plane strain (Supplementary Fig. 5). Therefore, the optical transition rate remains almost unchanged under in-plane uniaxial strain, leading to the insensitive responses of $\sigma^{yxx}$ to external strain (see Fig. 3d). For the $y$-polarized light, the optical transitions are mainly contributed by the coupling between anion $p$ states and Nb $d$ states. The energy gap between the $p$ and $d$ states retains under strain along the $x$ direction but increases when the strain along the $y$ direction changes from −2% to 2% (Supplementary Fig. 5c). As a result, the optical transition rate almost has no change under strain along $x$ direction but gradually decreases under strain along $y$ direction, in line with the calculated results that the $\sigma^{yyy}$ is insensitive to the strain along $x$ direction but sensitive to the strain along $y$ direction (see Fig. 3e). This unique strain-dependent shift current response, originating from the selective optical transitions between different orbitals that hold anisotropic deformation potentials to external strain[41], is quite different from other 2D material systems[42].

Besides, it is noteworthy that the $\sigma$ of NbOCl$_2$ is also independent of the sample thickness due to the unique crystal structure and weak interlayer coupling (see Fig. 3f). The strength of $\sigma$ peak is of the same order as first distinct peak of conventional FE BaTiO$_3$[27]. Compare to other 2D material like TMDs where NLO responses are suppressed in even-layer thickness[15], NbOCl$_2$ may have some advantages in realistic energy applications own to the layer-independent $\sigma$. Undoubtedly,

there are many other effects should be under consideration in realistic applications, like reported rectification in vdW NbOI$_2$ sheets[43].

## External-field-tunable NLO responses in NbO$X_2$

It is important to develop effective ways to control the NLO responses in NbO$X_2$. As proposed in Fig. 1b, the Nb atoms shifted along the $b$ direction can create a degree of freedom, which in principle may also generate different metastable AFE phases or PE phase with inversion symmetry that can effectively turn off the overall second-order NLO responses.

Whether any AFE phase can exist in NbO$X_2$ is still unknown in the experiments. In order to explore the possibility of realizing AFE phase in NbO$X_2$ along with the non-centrosymmetric/centrosymmetric control, we have synthesized the single crystals and performed X-ray structure analysis of NbOCl$_2$ and NbOI$_2$ as a function of external pressure. Differing from the scanning transmission electron microscopy that can only present the local atomic structure information, the single-crystal X-ray diffraction data can provide the global symmetry and the accurate bond information on polymorphic structures of NbO$X_2$[44]. Figure 4a shows the high-pressure experimental setup. At the ambient condition, the bulk NbOCl$_2$ exists the FE phase. As shown in Fig. 4b, the off-center displacements of Nb atoms along the $b$ direction cause them to reside on one side of the Cl-atom-formed plane (labeled as M$_{Cl}$ plane), generating a non-centrosymmetric polarized structure.

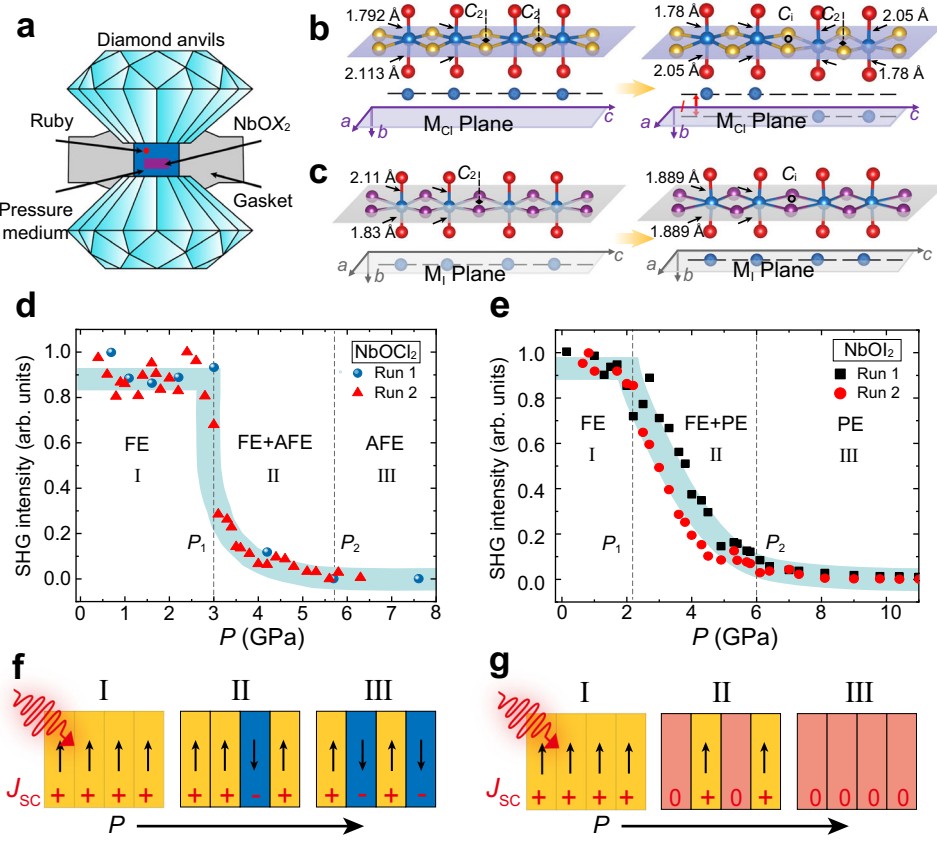

**Fig. 4 | Reversible FE-to-AFE (FE-to-PE) phase transition in NbOCl$_2$ (NbOI$_2$).** **a** Experimental high pressure setup, pressure is generated by two opposite diamond anvils. A gasket with a central hole is placed between two anvils to provide a chamber for samples and pressure transmitting medium. **b** Crystal structure of the FE NbOCl$_2$ at ambient pressure transforms to AFE phase at 5.7 GPa determined by our experiments. Schematic diagrams in the lower panel show the Nb atoms reside on from one side of the M$_{Cl}$ plane to both sides of the M$_{Cl}$ plane, with a distance $l$. **c** Crystal structure of the FE NbOI$_2$ at ambient pressure transforms to PE phase at

10.7 GPa. Schematic diagrams in the lower panel show the Nb atoms reside on from one side of the M$_I$ plane to on the M$_I$ plane. Corresponding structural parameters in Fig. 4b, c are marked in Supplementary Tables 2 and 12. **d** Normalized SHG intensity of NbOCl$_2$ as a function of external pressure (two different runs are performed under pressure). **e** same as **d** but for NbOI$_2$. **f, g** Schematic plot of local shift current ($J_{sc}$) generation for NbOCl$_2$ (NbOI$_2$) under pressure. Here, yellow (blue) rectangular represents one Nb–Nb dimer polarizes to +$b$ (−$b$) direction (*right* panel, Fig. 4b), and pink implies no polarization (*right* panel, Fig. 4c).

Interestingly, when we increase the pressure up to 5.7 GPa at room temperature, the collected single-crystal X-ray diffraction reveals that $NbOCl_2$ undergoes an unusual novel FE-to-AFE phase transition rather than simply to the centrosymmetric PE phase without local spontaneous polarization (Supplementary Fig. 6 for detailed single-crystal X-ray diffraction patterns and Supplementary Tables 1 and 2 for structural information). The structural analysis shows that the AFE structures maintain the vdW layered structure at 5.7 GPa. Comparing the AFE and FE structures, we find that the pressure mainly affects the Nb-O bond length but has fewer influences on the Nb-Cl bonds and interlayer coupling. We discover that the nearly linear Nb-O-Nb-O chains along the $b$ direction are almost constant, while the Nb-O bonds change with the pressure (1.78 Å and 2.05 Å at 5.7 GPa). More importantly, the off-center displacements of Nb atoms along the $b$ direction redistribute the Nb atoms on both sides of the $M_{Cl}$ plane, forming a centrosymmetric non-polarized structure with the twofold rotational symmetry retaining (space group of $C2/c$). In this AFE phase, every Nb pair composed of two adjacent Nb atoms shifts along different directions, separating them with the distance $l$ along the $b$ direction (0.373 Å at 5.7 GPa). The detailed crystallographic information and structural refinement parameters of $NbOCl_2$ measured in our experiments are listed in Supplementary Tables 2 and 6. We emphasize that the subtle structure changes cannot be well detected by powder X-ray diffraction patterns, e.g., the non-centrosymmetric space group $C2$ and the centrosymmetric space group $C2/c$ belong to the same diffraction group, and positions of their diffraction peaks are similar. Interestingly, this AFE phase has never been predicted. Our calculations also reveal that this AFE state is more stable than other metastable AFE states[45].

We note that the homogeneous structures with the existing in-plane AFE order are relatively rare, which so far are mainly found in inorganic or organic-inorganic perovskites[46,47], molecular crystals[48] and 2D vdW $In_2Se_3$[49]. Different from the $NbOCl_2$, the FE-to-PE can be detected in $NbOI_2$ under pressure and the single-crystal analysis under 10.7 GPa identifies a centrosymmetric structure with $C2/m$ space group, without local spontaneous polarization (Fig. 4c). The detailed single-crystal X-ray diffraction patterns are shown in Supplementary Fig. 7, and the crystallographic information and structural refinement parameters of $NbOI_2$ measured in our experiments are listed in Supplementary Tables 7–16.

The change of non-centrosymmetric structure to the inversion-symmetry-invariant structure also has important impacts on all the second-order NLO responses. Taking the SHG as an example, as shown in Fig. 4d, the normalized SHG intensity of single-crystal $NbOCl_2$ with transmission light $\lambda_{2\omega}$ of 532 nm is collected as a function of pressure under room temperature. Interestingly, the SHG intensity experiences three stages: (1) When the pressure is below ~3 GPa ($P < P_1$, stage I), the normalized SHG intensity remains constant, indicating the survival of FE phase at low pressure (Fig. 4b). (2) When the pressure is between ~3 and ~5.7 GPa ($P_1 < P < P_2$, stage II), the AFE phase starts to appear and mix with the FE phase. The higher the pressure, the larger the ratio of AFE phase over FE phase. Therefore, the SHG intensity continues to decrease. (3) When the applied pressure is above ~5.7 GPa ($P > P_2$, stage III), the FE sample is fully converted to the AFE one (Fig. 4b), finishing the phase transition. Eventually, the SHG intensity vanishes[50]. Even with the further increase of pressure, the SHG signal does not show up. We have also estimated the critical pressure for phase transition by comparing the total energy of FE and AFE phases under different pressures, which is consistent with our experiments. Besides external pressure, the temperature can cause a FE-to-AFE phase transition likewise in $NbOCl_2$ (Supplementary Fig. 8 and Supplementary Tables 17–21). In addition, to support that FE-to-AFE phase transition causes dramatical descent of SHG, the other possible scenarios, variation of band edge under external fields, have been ruled out (Supplementary Fig. 9).

In a similar way, we have performed the SHG measurements for $NbOI_2$ with varying pressure. As shown in Fig. 4e, the SHG intensity undergoes three stages into a PE phase, i.e., from FE phase (stage I) to FE–PE mixing phase (stage II) and to PE phase (stage III). The normalized SHG intensity starts to decrease at $P_1$ ~2 GPa and eventually reaches zero at $P_2$ ~6 GPa. When we release the pressure, the centrosymmetric phase becomes unstable and turns back to the FE one. Besides SHG, it is believed that the phase transition can also continuously tune the shift current responses. In addition, we note that it is a big challenge for us to perform variable wavelength SHG measurements, especially combined with diamond anvil cell to create a high-pressure environment.

Although both AFE and PE phase can give rise to a global zero second-order NLO response, the unique intralayer AFE phase may generate interesting hidden NLO responses. Here, taking BPVE effect as example, in BPVE theory the shift current arises from the difference of charge center in real space upon photoexcitation, which will reverse direction when the chirality is switched[51]. Therefore, although the AFE $NbOCl_2$ has a global centrosymmetric symmetry, the appearance of intralayer AFE nanostrips with local inversion-symmetry-breaking might generate an opposite flowing current while be zero macroscopically (stage III, Fig. 4f)[52]. The global shift-current intensity of $NbOCl_2$ under pressure depends on the ratio of mixed local AFE phase in the system (stage II), from a maximum value (stage I) to zero (stage III). On the other hand, this hidden BPVE cannot exists in the PE $NbOI_2$ phase that maintains the local inversion symmetry (stage III, Fig. 4g). Therefore, the current is exactly zero in the local PE phase region for $NbOI_2$ under pressure (stage II). The distinction may be helpful to provide an optical tool to probe the different AFE and PE phases, although both hold a global centrosymmetric symmetry. Therefore, the $NbOCl_2$ system may also provide an ideal platform for studying hidden second-order NLO effects, including both hidden photocurrent and hidden SHG.

## Discussion

It is interesting to understand the evolution of NLO properties as a function of $X$ in $NbOX_2$. As shown in Fig. 5a, the calculated SHG strengths for peak B in these three different $NbOX_2$ are nearly identical (see full SHG spectra in Supplementary Fig. 10a), consistent with the experimental observations[18,19]. For the optical transitions near the peak of $\chi^{(2)}_{yyy}$, we find that double-photon resonances in $NbOI_2$ and $NbOBr_2$ come from $X$ $p \to Nb$ $d$, while O also contributes in $NbOCl_2$ ($X$ $p +$ O $p \to Nb$ $d$) (Supplementary Fig. 11), resulting in an order of $NbOCl_2 > NbOI_2 > NbOBr_2$. Besides, a similar band nesting effect (Fig. 2e) exists in all these $NbOX_2$ but in different zones of $\boldsymbol{k}$ space (Supplementary Figs. 12 and 13), which is $X$-dependent. Together with other virtual energy terms, these two main factors integrated over the entire Brillouin zone, accidentally resulting in the strength of SHG being less sensitive to the different $X$.

Different from the SHG, the $\sigma$ is noticeably sensitive to the $X$ (see full $\sigma$ spectra in Supplementary Fig. 10b, c). As shown in Fig. 5b, the peak A of $\sigma^{yxx}$ [marked in Fig. 3a] increases with changing $X$ from Cl to I, but the peak B of $\sigma^{yyy}$ shows an opposite trend. To understand this unexpected phenomenon, we have calculated the $\boldsymbol{k}$-resolved optical transition and $\boldsymbol{k}$-resolved shift vector for these two peaks. For the $x$-polarized incident light, the optical transition is enhanced owing to the increase of $X$ orbitals in energy from Cl to I (see Fig. 5c, *upper* panel). Since the peak A of $\sigma^{yxx}$ is largely dominated by the $\varepsilon_2^{xx}$ (Fig. 3c), the enhancement of optical transition can increase values of peak A. On the other hand, for the $y$-polarized incident light, the shift vector gradually weakens when changing $X$ from Cl to I (see Fig. 5c, *bottom* panel). Because the $\sigma^{yyy}$ is mainly determined by the shift vector (Fig. 3b), the weakening of the shift vector results in the decrease of peak B.

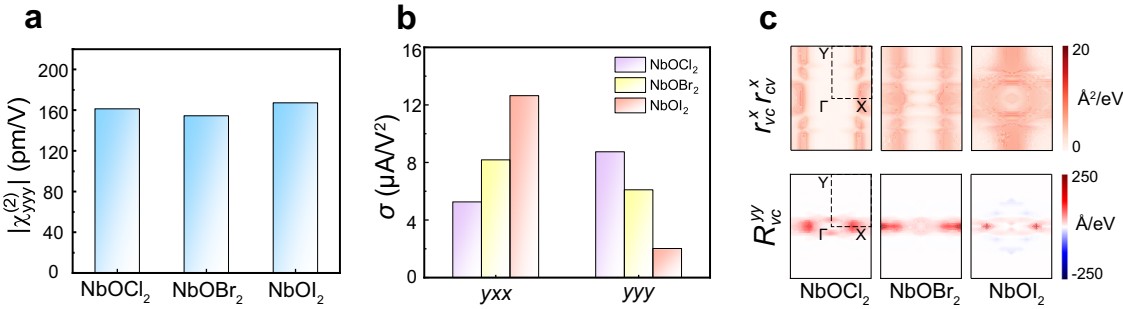

**Fig. 5 | NLO Properties of Monolayer NbO$X_2$. a** Intensity of peak B in NbO$X_2$ with $X$ changing from Cl to I. Feature of peak B is marked in Fig. 2b. **b** Intensities of peak A in $\sigma^{yxx}$ and peak B in $\sigma^{yyy}$ with $X$ changing from Cl to I in NbO$X_2$. Features of peaks A and B are marked in Fig. 3a. **c** $k$-resolved optical transition rate for peak A (*upper panel*) and shift vector for peak B (*bottom panel*) for monolayer NbO$X_2$.

In summary, combining theoretical and experimental studies, we present a systematical understanding of the manipulation of NLO properties in the family materials of NbO$X_2$. We propose a simple understanding of the origin of NLO responses, which may be helpful for further understanding and designing the various derivatives of NbO$X_2$. Using the advanced experiment technique, the single-crystal X-ray diffraction analysis technique under high pressure conditions, aided with extra SHG spectra and DFT calculations, we discover the intralayer FE-to-AFE phase transition in NbOCl$_2$ and FE-to-PE phase transition in NbOI$_2$ under certain pressures, which can in turn effectively manipulate the NLO properties in NbO$X_2$ as a result of crystal symmetry control, opening great opportunities for manipulating NbO$X_2$-based optoelectronic applications from optical sensing/computing/switch to BPVE. In addition, we propose that NbOCl$_2$ may be an interesting platform for studying hidden second-order NLO responses in the future.

## Methods
### Density functional theory calculations

Density functional theory (DFT)[53,54] calculations were performed using the Vienna ab initio simulation package (VASP)[55]. Here, the projector-augmented wave method (PAW)[56] was employed to treat the core electrons. The 520 eV energy cutoff for the plane-wave basis and the $\Gamma$-centered $5 \times 8 \times 1$ $k$-point mesh were used to ensure good convergence for the calculated results. A 20 Å vacuum was applied to avoid the interaction introduced by the periodic boundary conditions. For all the calculations, the Perdew–Burke–Ernzerhof (PBE) functional within the framework of generalized-gradient approximation[57] was used to treat the exchange-correlation term in the Kohn–Sham equation. For the band structure calculations, the Heyd–Scuseria–Ernzerhof (HSE06) hybrid functional[58] was adopted, which can correct the PBE-calculated bandgap closer to the experimentally measured one. Crystal structures were fully relaxed until the Hellmann–Feynman forces <0.015 eV/Å for each atom. The criterion for total energy convergence was set to be $1 \times 10^{-6}$ eV to ensure the accuracy of the results.

### NLO responses calculations

The NLO susceptibility tensors were calculated using our homemade package NOPSS, which was well tested and can be used to accurately calculate the SHG response[35,59], shift/injection currents, and photo-induced nonlinear spin current[60]. The SHG susceptibility tensor $\chi^{(2)}_{abc}(\omega)$ can be expressed as[61,62]

$$\chi^{(2)}_{abc}(-2\omega,\omega,\omega) = \chi^{abc}_{ter}(-2\omega,\omega,\omega) + \chi^{abc}_{tra}(-2\omega,\omega,\omega) \quad (3)$$

where the first term represents the SHG susceptibility tensor contributed by purely interband effects, and the second term originates from the mixing contribution of intraband and interband. Here, the

$\chi^{abc}_{ter}$ and $\chi^{abc}_{tra}$ were calculated by

$$\chi^{abc}_{ter}(-2\omega,\omega,\omega) = \frac{e^3}{\hbar^2\Omega} \sum_{nml,\mathbf{k}} \frac{r^a_{nm}\{r^b_{ml}r^c_{ln}\}}{(\omega_{ln}-\omega_{ml})}$$
$$\times \left[ \frac{2f_{nm}}{\omega_{mn}-2\omega} + \frac{f_{ln}}{\omega_{ln}-\omega} + \frac{f_{ml}}{\omega_{ml}-\omega} \right] \quad (4)$$

and

$$\chi^{abc}_{tra}(-2\omega,\omega,\omega) = \frac{i}{2}\frac{e^3}{\hbar^2\Omega}\sum_{nm,\mathbf{k}} f_{nm}\left[ \frac{2}{\omega_{mn}(\omega_{mn}-2\omega)} r^a_{nm}\left(r^b_{mn;c}+r^c_{mn;b}\right) \right.$$
$$+ \frac{1}{\omega_{mn}(\omega_{mn}-\omega)}(r^a_{nm;c}r^b_{mn}+r^a_{nm;b}r^c_{mn})$$
$$+ \frac{1}{\omega^2_{mn}}\left(\frac{1}{\omega_{mn}-\omega}-\frac{4}{\omega_{mn}-2\omega}\right)r^a_{nm}\left(r^b_{mn}\Delta^c_{mn}+r^c_{mn}\Delta^b_{mn}\right)$$
$$\left. - \frac{1}{2\omega_{mn}(\omega_{mn}-\omega)}(r^b_{nm;a}r^c_{mn}+r^c_{nm;a}r^b_{mn}) \right] \quad (5)$$

The shift current tensor $\sigma$ was calculated using the formula[30]

$$\sigma^{abc}(0;\omega,-\omega) = -\frac{i\pi e^3}{2\hbar^2}\int\frac{d\mathbf{k}}{8\pi^3}\sum_{nm}f_{nm}\left(r^b_{mn}r^c_{nm;a}+r^c_{mn}r^b_{nm;a}\right)\delta(\omega_{mn}-\omega) \quad (6)$$

When the light is limited to polarize linearly in the $b$ direction, it can also be expressed more concisely as

$$\sigma^{abb}(0;\omega,-\omega) = \frac{\pi e^3}{\hbar^2}\int\frac{d\mathbf{k}}{8\pi^3}\sum_{n,m}f_{nm}R^{ab}_{nm}\left|r^b_{nm}\right|^2\delta(\omega_{mn}-\omega) \quad (7)$$

where the shift vector is defined by

$$R^{ab}_{nm}(\mathbf{k}) = \frac{\partial\phi^b_{nm}(\mathbf{k})}{\partial k^a} - \xi^a_{nn}(\mathbf{k}) + \xi^a_{mm}(\mathbf{k}) \quad (8)$$

To explore the origin of $\sigma$, the integrated shift vector $e\bar{R}^{ab}$ and linear optical absorption spectrum $\varepsilon^{ab}_2$ were employed, which are given by[40]

$$e\bar{R}^{ab} = e\Omega\int\frac{d\mathbf{k}}{8\pi^3}\sum_{n,m}f_{nm}R^{ab}_{nm}\delta(\omega_{nm}-\omega) \quad (9)$$

and

$$\varepsilon^{ab}_2(-\omega;\omega) = \frac{\pi e^2}{\hbar}\int\frac{d\mathbf{k}}{8\pi^3}\sum_{n,m}f_{nm}r^a_{nm}r^b_{mn}\delta(\omega_{mn}-\omega) \quad (10)$$

Among Eqs. (3)–(10), the connection $\boldsymbol{\xi}_{nm} = \frac{i(2\pi)^3}{\Omega} \int d\boldsymbol{r}\, u_n^*(\boldsymbol{k},\boldsymbol{r}) \nabla_{\boldsymbol{k}} u_m(\boldsymbol{k},\boldsymbol{r})$. If $n \neq m$, $r_{nm} = \boldsymbol{\xi}_{nm}$, otherwise $r_{nm} = 0$, where $\Omega$ is the volume of the unit cell, $u_m(\boldsymbol{k},\boldsymbol{r})$ is the periodic part of Bloch wavefunction. Superscripts $\{n, m, l\}$ and $\{a, b, c\}$ represent the band indices and Cartesian indices, respectively. $\{r_{ml}^b r_{ln}^c\} = \frac{1}{2}(r_{ml}^b r_{ln}^c + r_{ml}^c r_{ln}^b)$. $f_{nm} = f_n - f_m$, where $f_n(f_m)$ is the Fermi distribution function. $\hbar\omega_{nm} = \hbar\omega_n - \hbar\omega_m$, where $\hbar\omega_n(\hbar\omega_m)$ is energy. The generalized derivative of the dipole matrix element $r_{nm;a}^b$ was expressed as

$$r_{nm;b}^a(\boldsymbol{k}) \equiv \frac{\partial r_{nm}^a(\boldsymbol{k})}{\partial k^b} - i\left[\xi_{nn}^b(\boldsymbol{k}) - \xi_{mm}^b(\boldsymbol{k})\right] r_{nm}^a(\boldsymbol{k}) \quad (11)$$

By using the sum rule of the generalized derivative of the dipole matrix element, we can rewrite this term as

$$r_{nm;a}^b = \frac{r_{nm}^a \Delta_{mn}^b + r_{nm}^b \Delta_{mn}^a}{\omega_{nm}} + \frac{i}{\omega_{nm}} \times \sum_l \omega_{lm} r_{nl}^a r_{lm}^b - \omega_{nl} r_{nl}^b r_{lm}^a \quad (12)$$

which is equivalent to the original formula and is more convenient for numerical calculation (Supplementary Note 2). $\Delta_{mn}^b = \upsilon_{mm}^b - \upsilon_{nn}^b$ represents the difference in the electron group velocities. $\phi_{nm}^b$ is the phase of $r_{nm}^b = |r_{nm}^b|e^{i\phi_{nm}^b}$ and the shift vector follows the relationship $R_{nm}^{ab} = \frac{\mathrm{Im}\, r_{nm}^b r_{mn;a}^b}{r_{nm}^b r_{mn}^b}$.

Here we employ the scissor operator from HSE06 functional calculations to get an accurate band gap. The dense of $k$-point mesh and the number of bands were carefully examined for convergence, and the $19 \times 32 \times 1$ k-point mesh and 120 electronic bands were finally chosen to perform calculations (Supplementary Fig. 14). For all calculations, 200 grids were linearly interpolated in the energy range of 0–6 eV to make the curve smooth, and the small imaginary smearing factor appeared in the susceptibility tensors expression was set to be $\eta = 0.05$ eV, a common value used for many 2D NLO material calculations[17,35]. In addition, to compare the NLO effects of systems with different thicknesses, the effective susceptibility was employed, which was defined as $\chi_{\mathrm{eff}}^{\mathrm{NLO}} = \frac{\chi^{\mathrm{NLO}*}L_s}{L_z}$, where $L_s$ is the thickness of the slab model and $L_z$ is the thickness of 2D crystal plus the vdW thickness of bulk NbO$X_2$. In experiment[19], the SHG intensity is characterized by the intensity of the outgoing light ($I$). The relationship between the SHG intensity measured in experiments and the SHG strength discussed in this paper can be expressed as $I \propto |P_{NL}(2\omega)|^2 \propto |N\chi_{\mathrm{monolayer}}^{(2)}|^2$.

## Crystal growth

Chemical vapor transport was used to synthesize NbOCl$_2$ single crystals, as reported previously[19]. Niobium pentoxide powder (99.99% purity), niobium power (99.9% purity), and niobium pentachloride powder (99.999% purity) were mixed stoichiometrically and sealed under vacuum ($10^{-6}$ torr) in an evacuated quartz tube. The quartz tube was placed in a 2-zone furnace, where the temperature of volatilization and crystallization regions were set to 500–600 °C and 300–400 °C with a growth period of 5 days. In the end, NbOCl$_2$ single crystals, rectangle-shaped thin plates, could be collected in the crystal growth region. NbOI$_2$ single crystals were grown by the chemical vapor transport method too. Niobium powder (99.95% purity), Nb$_2$O$_5$ powder (99.99% purity), and iodine flakes (99.999% purity) in a 3:1:10 molar ratio were put into a silicon tube with a length of 200 mm and an inner diameter of 14 mm. The tube was pumped down to 0.01 Pa, sealed under vacuum, and then placed in a two-zone horizontal tube furnace. The two growth zones are raised slowly to 1083 and 943 K for 3 days and are then held there for another 5 days. Shiny, platelike crystals with lateral dimensions of up to several millimeters can be obtained from the growth (Supplementary Fig. 15). In order to avoid degradation, the NbOI$_2$ crystals are stored in an Ar-filled glovebox.

## Experimental setup

High-pressure experiments were carried out in a Mao-Bell type diamond anvil cell (DAC) with 300 μm culet anvils and steel gaskets. Neon was loaded as the pressure-transmitting medium. Ruby fluorescence is used for the pressure measurement[63]. The DAC was mounted and centered on a Bruker D8 Venture four-circle diffractometer according to Dawson et al.[64]. Single crystal X-ray diffraction data were collected at some specific pressures and temperatures with multilayer monochromator Mo Kα radiation ($\lambda = 0.71073$ Å). Data were collected in ω-scans in eight settings of 2θ and φ with a frame and step size of 60 s and 0.5°, respectively. The data collection strategy was based on that described by Dawson et al.[64]. The sample reflections were harvested manually by using the SMART code. Image masks, to avoid integrating the signal from detector regions obscured by DAC, were created using the program ECLIPSE[65]. For high-temperature single crystal x-ray diffraction data, a black block-shaped single crystal of NbOCl$_2$ with dimensions of $0.063 \times 0.043 \times 0.039$ mm$^3$ was selected and fixed on the top of a thin glass fiber. X-ray diffraction data were collected by increasing the temperature to 500(2) K using an Oxford Cryosystems Plus low-temperature device and were measured on a Bruker D8 Venture four-circle diffractometer with multilayer monochromator Mo Kα radiation ($\lambda = 0.71073$ Å). Data integration and oblique correction were performed with the software package SAINT[66]. Absorption effects were corrected using the Multi-Scan method (SADABS)[67]. Structures were solved by dual space methods (SHELXT) and refined by full-matrix least-squares on $F^2$ (SHELXL)[68] using the graphical user interface ShelXle[69]. All atoms were refined anisotropically for the room-pressure and the high-pressure structures.

## Data availability

The data that support the findings of this study are available from the corresponding author upon request.

## Code availability

The NOPSS code that supports the findings of this study is available from the corresponding author upon reasonable request.

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

## Acknowledgements

This work is supported by the NSFC (Grant No. 12088101), National Key Research and Development of China (Grant Nos. 2022YFA1402401, 2018YFE0202600, 2022YFA1403800, and 2022YFE0109200), the National Science Fund for Distinguished Young Scholars (Grant No. T2225027), NSAF (Grant No. U2230402), and the Beijing Natural Science Foundation (Grant No. Z200005). The calculations were done in the Tianhe-JK cluster at CSRC. Some experiments are supported by the Synergic Extreme Condition User Facility.

## Author contributions

B.H. and H.G. conceived the overall project. L.T.Y. performed all the calculations. X.J. developed the computational code for NLO calculations and helped with the calculations. Q.G., X.C., S.Y., X.Y.W., X.Q.W., X.Z. and H.L. synthesized the samples. W.Z., D.H.J. and H.G. performed the single crystal analysis. D.H., D.Q.J., Y.W. and H.G. performed the experimental SHG measurements. L.T.Y., Y.L., W. Z. H.G. and B.H. analyzed all the data. L.T.Y., Y.L., W. Z., H.G. and B.H. wrote the paper with input from all other authors. All the authors discussed and contributed to the paper.

## Competing interests

The authors declare no competing interests.
