## [Peer Review File · Nature Communications]

Manipulation of Nonlinear Optical Responses in Layered Ferroelectric Niobium Oxide DihalidesEditorial Note: Parts of this Peer Review File have been redacted as indicated to remove third-party material where no permission to publish could be obtained.

REVIEWER COMMENTS

Reviewer #1 (Remarks to the Author):

The authors report nonlinear optics of Niobium oxide dihalides. The results are interesting.

- (i) Can the authors upload their codes to the GitHub repository so that the readers can check the codes and others?
- (ii) can the authors calculate Fig.2 for all three materials?
- (iii) In fig. 1b, what is the estimated external field to introduce the transition?
- (iv) Can the wavelength dependence measurement be carried out to confirm the theoretical estimation in Fig. 2b/f?
- (v) Fig. 4 results are beautiful. Can the authors carry out the experiments with the other 2 materials?

Reviewer #2 (Remarks to the Author):

Following the recent experimental work (Nature 613, 53–59 (2023)), authors performed simulations to explore the origin of large SHG strength in two-dimensional NbOCl₂ and NbOI₂ compounds. Besides, high-pressure measurements are also performed to tune the non-linear optical responses of NbOCl₂ crystal. The current work is more like a continuation of previous publication, without producing sufficient research novelty and physical insights. My detailed comments and questions are summarized below:

1. The authors performed first-principles calculations and claimed that the large SHG in NbOCl₂ is dominated by the synergy between a large transition dipole moment and a band-nesting-induced large intensity of electron-hole pairs. However, this conclusion is not very meaningful since the above two dominant factors may work for any system with large SHG response. Furthermore, the nonlinear optical responses are not solely determined by electronic structure but also related to the geometric properties of Bloch electrons. The author used a sum rule of the generalized derivative of the dipole matrix element and these geometric effects are not considered.
2. The maximum second-harmonic generation (SHG) susceptibility of NbOCl₂ authors simulated is around 120 pm/V. While the experimental measured value is around 200 pm/V (Fig. 3 (g) of Ref. 19). Authors should reproduce the experimental results before they could explain the origin of large SHG response.
3. The authors used a “trick” on the experimental data, as shown in Fig. 2f. They narrowed the range of the experimental data (SHG susceptibility) by 1/4 to make the experimental data fit the calculation well. This is not a scientific way to present raw data.
4. SHG measurement on NbOBr₂ has never been conducted by experiment yet, “niobium oxide dihalides NbOX₂ (X= Cl, Br, I) has attracted plenty of interests due to their large, anisotropic and even layer-dependent SHG responses” is not a valid statement.

5. Bulk NbOX₂ contains two monolayers in its primary unit (Ref. 22-24), so it is unlikely to displaying ABC stacking along out-of-plane z axis.

6. Authors perform the symmetry derived SHG susceptibility analysis on NbOCl₂ monolayer with Orthorhombic C_{2v} point group symmetry. While experimental measurements are performed on NbOCl₂ nanoflakes with multiple layers. Bulk NbOCl₂ adopts C₂ symmetry, such a monoclinic symmetry will also work for NbOCl₂ multiple layers. Compared to Orthorhombic C_{2v} symmetry, monoclinic C₂ symmetry without mirror plane has completely different symmetry selection rules. In other words, are SHG simulations performed on C_{2v} NbOCl₂ monolayer comparable to monoclinic NbOCl₂ layers?

7. In page 8, "SHG strength of NbOCl₂ is almost independent of the sample thickness". In Fig. 3 e of Ref. 19, the significant thickness dependent SHG intensity is observed in NbOCl₂ nanoflakes. What is the different between "SHG strength" in current work and "SHG intensity" from Ref. 19?

8. In Page 10, authors mentioned "NbOCl₂ a promising material in the field of energy conversion". In fact, ferroelectric photovoltaic effect of NbOI₂ nanoflakes has been measured by experiment (Appl. Phys. Lett. 119, 033103 (2021)). Even NbOI₂ has a much smaller band gap, it still displays a quite weak photocurrent. Compared to NbOI₂, NbOCl₂ has larger band gap and therefore is more insulating. Is insulating NbOCl₂ really promising for solar energy conversion?

9. Based on my knowledge of X-Ray diffraction measurement, I did not see any FE-to-AFE phase or symmetry change induced diffraction peak change. How can authors conclude that there is a FE-to-AFE phase transition in NbOCl₂ single crystal sample based on X-Ray diffraction measurement?

10. Measurement of NbOCl₂ crystal under pressure up to 10 GPa, has been conducted in Fig. S20 of Ref. 19. NbOCl₂ is known for its significant optical nonlinearity and large SHG response. In current work, authors reported SHG responses of NbOCl₂ crystal can be completely suppressed under the pressure around 5.7 GPa. So, from high SHG response to zero SHG response. Is pressure really an effective way to "fine tune" nonlinear optical responses of NbOCl₂?

11. Antiferroelectric (AFE with ↑↑↓↓ polarization arrangement) phase reported in current work is more like two short-periodic FE single domain with opposite polarization, separated by 180 FE domain wall (Fig. S11 of Ref. 18). While each single FE domain should still have non-zero and detectable SHG signal. Based on authors logic, ↑↑↓↓ AFE phase is more stable than ↑↓ AFE phase, will ↑↑↑↓↓↓ or ↑↑↑↑↓↓↓ AFE phase also exist in NbOCl₂ crystal under pressure?

Reviewer #3 (Remarks to the Author):

The manuscript theoretically investigates the band structures of niobium oxide dihalides and their optical properties such as SHG coefficients, absorption strength, shift current coefficients. The calculation gives insight of these optical properties such as the origin of the optical features. In addition, the authors claimed to experimentally observe the antiferroelectric (AFE) phase and demonstrate how to tune from ferroelectric (FE) phase to AFE phase. Overall, the calculation and explanation of the optical features are insightful.

However, the most interesting part to Nature Communications, the claim of AFE phase, is lack of solid evidence in this manuscript. A few issues should be addressed for reconsideration for publication.

1. X-ray diffraction should be direct evidence to reflect the structures of AFE and FE phases. In this work, the authors measured the X-ray diffraction at 500 K and at 5.7 GPa, respectively. However, only one figure S5 in the supplementary information to show the pattern does not justify differentiating the subtle change between AFE and FE. I suggest the authors show the theta-scan figures and the fitting results of the key diffraction peaks. For example, by Comparing with 300 K and 500K (or with atmosphere pressure and 5.7 GPa), could the authors clearly identify the difference between the key peaks corresponding to the AFE and FE phase? A comparison between the AFE and FE diffraction data will help to clarify if the subtle change is in the error range for fitting.

2. The authors measured the second harmonic generation (SHG) of NbOCl₂ as a function of pressure and temperature. While the crystal was under high pressure or temperature, the SHG efficiently dramatically dropped. The authors attributed the diminishing SHG to AFE with inversion symmetry. Other possible scenarios should be ruled out to support this scenario as an indirect evidence. For example, in Fig. 2f, the authors show the SHG coefficient spectra, which exhibit an edge around 2 eV. SHG intensity is sensitive for wavelengths around the edge. Will the SHG intensity vary due to the shift of the edge (or shift of the bandgap corresponding to B peak) for different temperature or pressure? In other words, is there dependence of temperature or pressure on the SHG coefficient spectra?

3. Although the authors calculated that the total energy of AFE is only 2meV/u.c higher than that of FE as shown in Fig. 4f, it is not straightforward to me that over a certain temperature or pressure, the whole crystals would be completely transformed into a slightly higher energy AFE state, rather than co-existence of AFE and FE states in random order. Although temperature and pressure provide a driving force to overcome the barrier energy, they do not specifically drive the system toward AFE state. Typically for ferroelectric or ferromagnetic materials, the ferroic property disappears above a transition temperature (Curie temperature). Could it be possible that NbOCl₂ simply becomes centrosymmetric without local spontaneous polarization above a certain temperature or pressure?

4. For the band structure calculation, the authors mentioned a scissor operation is used. The authors should show how much energy is shifted according to the measured bandgap from which literature. Or it would be better that the authors experimentally determine the bandgap of their crystals. For example, in Fig. 2f, the experimental data from Ref. 19 was blue-shifted 0.6 eV for comparison. Does the bandgap already shift to the right energy referred to Ref. 19 while a 0.6 eV shift is still necessary for comparing SHG coefficient spectra?

5. In the introduction “Among numerous materials, compounds without inversion symmetry can exhibit various nonlinear optical (NLO) responses” might require revision. NLO responses also include third order nonlinearity, which exists all kinds of materials. The authors might mean second-order nonlinear responses rather than NLO responses.

6. In page 7 “Again, this strong anisotropic feature is absent in many existing 2D materials” might be wrong. Most 2D material crystals exhibit anisotropic features if they are non-centrosymmetric. The angle-resolved SHG pattern reflects the group symmetry of the

crystal. Graphene and Bi₂Se₃ in Ref. 20, 21 exhibit inversion symmetry. The SHG originates from the surface or interface where the symmetry breaks.

Response Letter

We thank the referees for their careful review on the manuscript, which is very helpful to improve the quality of our manuscript. We have carefully considered all the comments and made responses as follows. Meanwhile, the changes in the manuscript are marked as blue color.

Response for the first referee:

Comment 1: “The authors report nonlinear optics of niobium oxide dihalides. The results are interesting.”

Response 1: We thank the referee for his/her positive comment on our manuscript.

Comment 2: “Can the authors upload their codes to the GitHub repository so that the readers can check the codes and others?”

Response 2: We thank the referee for this good suggestion, which actually is our ultimate goal for developing this simulation package. Unfortunately, the current version is a primitive version without integrating different functionalities and without user manual. We are now working hard to improve our codes, such as extending important functionalities, improving computing efficiency, writing detailed user manual. We will open the source code later when it has a relatively good shape.

To eliminate the concerns of the referee on the reliability of our codes, we have performed calculations on two typical NLO materials for benchmark. Figure R1 shows the calculated SHG and shift-current of two different monolayer materials (MoS₂ and SnSe) using our code, in comparison with the results reported in literatures from other groups [Nano Lett. 17, 5027–5034 (2017); Sci. Adv. 5, eaav9743 (2019)]. As one can see, we can obtain almost the same results as others, which proves the reliability of our codes.

Revision: (1) We have added the sentence “*The NLO susceptibility tensors were calculated using our homemade package NOPSS, which was well tested and can be used to accurately calculate the SHG response [35,59], shift/injection currents, and photo-induced nonlinear spin current [60].*” in Page 18. (2) We have added the statement of “*The NOPSS code that supports the findings of this study is available from the corresponding author upon reasonable request.*” in the Code availability section.

[Redacted]

Fig. R1. **a** Comparison of our calculated SHG of monolayer MoS₂ (*left panel*) and the reported results (*right panel*). Here, the reference is reprinted from the Fig. 2 in Nano Lett. **17**, 5027–5034 (2017). **b** Comparison of our calculated shift-current susceptibility component of monolayer SnSe (*left panel*) and the reported result (*right panel*). Here, the reference is reprinted from the Fig. S1 in supplementary information of Sci. Adv. **5**, eaav9743 (2019).

Comment 3: “Can the authors calculate Fig. 2 for all three materials?”

Response 3: Following the referee’s suggestion, we have calculated the related data for NbOBr₂ and NbOI₂, and the results are shown in Fig. R2 for NbOBr₂ and Fig. R3 for NbOI₂. As one can see, NbOBr₂ and NbOI₂ establish the similar physical mechanism of nonlinear optical responses as that of NbOCl₂ discussed in the main text.

Fig. R2. **a** Calculated nonlinear susceptibility $|\chi_{abc}^{(2)}|$ of monolayer NbOBr₂ as a function of incident

photon energy. **b** SHG response for monolayer, bilayer, and bulk NbOBr₂. **c** *k*-resolved absorption strength, $r_{mn}^y r_{nm}^y$ (Å²), occurred near A and B peaks in the SHG spectrum. Here, VB_{*n*} (CB_{*n*}) labels the *n*th valance (conduction) band starting from the top valance (bottom conduction) band. **d** Contour plot of energy difference of specified two bands. **e** Angle-resolved SHG polarization. Here, the photon energy of the incident light is set to be 1.96 eV, and the red (blue) curve represents the response with the direction perpendicular (parallel) to the polarization of incident light.

Fig. R3. **a** Calculated nonlinear susceptibility $|\chi_{abc}^{(2)}|$ of monolayer NbOI₂ as a function of incident photon energy. **b** SHG response for monolayer, bilayer, and bulk NbOI₂. **c** *k*-resolved absorption strength, $r_{mn}^y r_{nm}^y$ (Å²), occurred near A and B peaks in the SHG spectrum. Here, VB_{*n*} (CB_{*n*}) labels the *n*th valance (conduction) band starting from the top valance (bottom conduction) band. **d** Contour plot of energy difference of specified two bands. **e** Angle-resolved SHG polarization. Here, the photon energy of the incident light is set to be 1.69 eV, and the red (blue) curve represents the response with the direction perpendicular (parallel) to the polarization of incident light.

Revision: (1) We add the sentence “Besides, the similar band nesting effect (Fig. 2e) exists in all these NbOX₂ but in different zones of *k* space (Supplementary Figs. 12 and 13), which is *X* dependent” in page 16. (2) We have replotted Supplementary Fig. 11 and added Figs. R2-R3 as Supplementary Figs. 12-13.

Comment 4: “In Fig. 1b, what is the estimated external field to introduce the transition?”

Response 4: We thank the referee for asking this good question. In this study, the

external field used to create phase transition is pressure or temperature. In our experiments, the single-crystal X-ray diffraction data and SHG measurement reveal that the phase transition is fully realized when the temperature reaches 450 K or the pressure reaches 5.7 GPa in NbOCl₂.

It is always a challenge to accurately estimate the critical strength of external field for phase transition in theory. The FE-to-AFE phase transition may happen when the energy caused by temperature can suppress the energy barrier between the FE and AFE phases. In a previous work on a similar material system [Nanoscale Horiz. **4**, 1113 (2019)], the energy barrier of ~14 meV/f.u. is corresponding to a critical temperature of ~400 K using Monte Carlo simulation. Therefore, the energy barrier of ~19 meV/f.u. in our current study is corresponding to a slightly higher temperature than 400 K, consistent with the experimental observation.

To estimate the critical pressure, we have calculated the total energy of the FE and AFE phases of NbOCl₂ under different pressures. As shown in Table R1, we find that the total energy of the FE phase is lower than that of the AFE phase under 3 GPa, while the AFE phase gets the lower total energy at 6 GPa. Therefore, the critical pressure should be within the region of 3 - 6 GPa, consistent with the experimental observation.

We must note that the real phase-transition path is too complex to be accurately estimated. For example, the real transition state is a mixed state of random FE and AFE domains, which may not be described using a specific crystal structure in the calculations. That's why we believe the experimental data presented in our current manuscript is important to guild future studies on the phase transitions in this material system.

Revision: We have added the following discussion “*We also have estimated the critical pressure for phase transition by comparing the total energy of FE and AFE phases under different pressures, which is consistent with our experiments.*” in Page 13.

Comment 5: “Can the wavelength dependence measurement be carried out to confirm the theoretical estimation in Fig. 2b/f?”

Response 5: We thank the referee for this very nice suggestion. The wavelength-dependent measurement method is usually used to characterize structures in a combination with low temperature. To be honest, it is a big challenge for us to perform variable wavelength SHG measurements, especially combined with diamond anvil cell to create a high-pressure environment. We regret that we cannot directly supplement such data.

On the other hand, our experiments already have provided the direct evidence for the presence and the manipulation of second-order nonlinear effects, as illustrated by the single-wavelength test.

Revision: We have added the following discussion “*We note that it is a big challenge for us to perform variable wavelength SHG measurements, especially combined with diamond anvil cell to create a high-pressure environment.*” in Page 14.

Comment 6: “Fig. 4 results are beautiful. Can the authors carry out the experiments with the other 2 materials?”

Response 6: Following the referee’s suggestion, we have synthesized bulk NbOI₂ through conventional Chemical Vapor Transport (CVT) method after many hard attempts and then performed high-pressure single-crystal analysis and SHG measurements. As shown in Fig. R4, SHG measurements show that the SHG intensity became weak starting from ~2 GPa and completely loss above ~6 GPa, indicating a structural transition of NbOI₂ under pressure.

The single-crystal X-ray analysis has been further performed under pressure to understand the new high-pressure phase. Different from NbOCl₂, the single-crystal analysis identifies the high-symmetric paraelectric (PE) structure with *C2/m* space group under high pressure, instead of the AFE phase with *C2/c* space group found in NbOCl₂. The detailed information of this new structure can be found in Supplementary Tables 6-10.

Furthermore, to explore whether any possible AFE phases exist under pressure, we have performed theoretical calculations on NbOI₂ under 6 and 10 GPa. Different from NbOCl₂, we find that the FE phase of NbOI₂ at 0 GPa will transform into the high-symmetric PE structure under both 6 GPa and 10 GPa after structural optimization, which is consistent with our experiment. It implies that the AFE phase is indeed unstable under pressure for NbOI₂. Therefore, we conclude that pressure can induce a FE-to-AFE transition in NbOCl₂ but a FE-to-PE transition for NbOI₂.

Despite many attempts, it is still challenging for us to synthesize high-quality NbOBr₂ samples in a short time. We apologize that we cannot provide the experimental data for NbOBr₂ at this time. Instead, we have performed the calculations on NbOBr₂. Same as NbOI₂, NbOBr₂ transforms into a high-symmetric PE structure without localized polarization at 6 GPa. Therefore, we speculate NbOBr₂ also cannot establish the FE-to-AFE phase transition. In conclusion, we can expect that NbOCl₂ is unique in the NbOX₂ family that can realize a FE-to-AFE transition under pressure. We have put all the structural details of NbOCl₂ and NbOI₂ in supplementary information Tables. S1-S10 for other researchers to repeat or use our results.

Revision: (1) We have added Fig. R4 as Fig. 4d in the main text. (2) We have summarized the structural information of NbOI₂ into the Tables 6-10 in supplementary information. (3) We have added the growth method for NbOI₂ in the Method section.

Fig. R4. Normalized SHG intensity of NbOI₂ as a function of external pressure (two different runs are performed under pressure).

Response for the second referee:

Comment 1: “Following the recent experimental work (Nature 613, 53-59 (2023)), authors performed simulations to explore the origin of large SHG strength in two-dimensional NbOCl₂ and NbOI₂ compounds. Besides, high-pressure measurements are also performed to tune the non-linear optical responses of NbOCl₂ crystal. The current work is more like a continuation of previous publication, without producing sufficient research novelty and physical insights.”

Response 1: We thank the referee for his/her careful review on our manuscript. Indeed, our work is related to the previous work in Nature. However, our work reported many new results that are not discovered previously, as clarified in the following:

Significance of physical mechanism: The vdW layered crystals NbOCl₂ and NbOI₂ establish unexpected giant and layer-independent second-harmonic generation (SHG), different from many other widely studied 2D layered NLO materials (such as TMDs). These unique NLO responses suggest the great potential of NbOCl₂ and NbOI₂ on designing advanced light source. *Our work proposes a simple understanding on the origin of novel NLO responses of niobium oxide dihalides, which may be helpful for further understanding and designing the various derivatives of niobium oxide dihalides.*

Importance of physical phenomenon: Beyond the previous works, we have revealed a phase transition from ferroelectric (FE) phase to antiferroelectric (AFE) phase when applying pressure or temperature, which is nontrivial. *Firstly, the pressure or temperature driven FE-to-AFE phase transition is rare.* In general, increasing temperature may introduce strong thermal fluctuation to destroy long-range FE order, resulting in paraelectric (PE) phase instead of the ordered AFE phase in most materials [e.g., see Nat. Rev. Mater. 2, 16087 (2017); Nat. Commun. 12, 152 (2021)]. On the other hand, increasing pressure may suppress the polarization [e.g., see Phys. Rev. Lett. 35, 1767 (1975); Phys. Rev. B 89, 205122 (2014)]. Therefore, it is very nontrivial for discovering the FE-to-AFE phase transition driven by moderate pressure (5.7 GPa) or temperature (450 K). *Secondly, the phase transition from FE phase to intralayer AFE*

phase in vdW layered materials is also very rare. There is only a few vdW layered materials that can establish the FE-to-AFE phase transition, and most of them are transformed from FE phase to the *interlayer* (not intralayer) AFE phase. To the best of our knowledge, the NbOCl₂ is a new example which is experimentally found to establish the intralayer AFE phase under suitable conditions except for In₂Se₃ [Phys. Rev. Lett. 125, 047601 (2020)].

Irreplaceability of experiment technique: Finally, we would like to mention that the *experiment technique used in our work, i.e., the single-crystal X-ray diffraction analysis technique under high pressure condition, is also nontrivial.* As shown in Figs. 4b and 4c in the main text, the FE and AFE phases differ in the displacement of Nb atoms along the *b* direction. This small difference cannot be well distinguished by powder X-ray diffraction analysis and Raman spectrum measurement, i.e., the FE and AFE phases establish the almost same the powder X-ray diffraction spectrum and Raman spectrum (see detailed discussion in **Response 9**). Compared to the standard methods, the single-crystal X-ray diffraction analysis technique adopted in our work can provide precise structural information, consistent with our SHG measurement. This is the irreplaceability of experiment technique, especially for revealing the small structural changes under high-pressure condition, as the case of NbOX₂.

In summary, we believe that our work can provide sufficient research novelty and physical insights from different aspects, which is suitable for publishing in Nature Communications.

Revision: We have added the above discussion in Pages 17 (summary section) to highlight the novelty of our study.

Comment 2: “The authors performed first-principles calculations and claimed that the large SHG in NbOCl₂ is dominated by the synergy between a large transition dipole moment and a band-nesting-induced large intensity of electron-hole pairs. However, this conclusion is not very meaningful since the above two dominant factors may work for any system with large SHG response. Furthermore, the nonlinear optical responses are not solely determined by electronic structure but also related to the geometric

properties of Bloch electrons. The author used a sum rule of the generalized derivative of the dipole matrix element and these geometric effects are not considered.”

Response 2: We thank the referee for raising this important question. We agree with the referee that SHG is strongly associated with the linear optical response, which is mainly determined by transition dipole moment and joint electron-hole density of states (e.g., see Appl. Phys. Lett. 5, 17 (1964); Phys. Rev. B 102, 045411 (2020)). In our study, we can clearly demonstrate that the origin of large joint density of states are paritally from band nesting effect in different zones of k space for different NbOX₂ (also see Figs. R2 and R3), providing a clear band origin. Meanwhile, for the second-order shift current, we can clearly identify the peaks in NbOX₂ are mainly originated from the integration of shift vector or from the transition probability (see Fig. 3 in the main text). Finally, we can give an overall understanding on the evolution of SHG or shift-current peaks as a function of X in NbOX₂ (see Fig. 5).

In the following, we will make a response to the comment of geometric properties from our point of view as much as possible.

As the referee pointed out, the NLO responses are related to the geometric properties of Bloch electrons. For example, the generalized derivatives of matrix elements can be expressed in geometric form,

$$r_{nm;b}^a(\mathbf{k}) \equiv \frac{\partial r_{nm}^a(\mathbf{k})}{\partial k^b} - i[\xi_{nn}^b(\mathbf{k}) - \xi_{mm}^b(\mathbf{k})]r_{nm}^a(\mathbf{k}). \quad (1)$$

Here, a, b indicate Cartesian components, $\xi_{nm}^a = i \int dx u_n^*(\mathbf{k}, \mathbf{x}) \frac{\partial}{\partial k^a} u_m(\mathbf{k}, \mathbf{x})$, which is called Berry connection when $n = m$, and $r_{nm}^a(\mathbf{k}) \equiv \xi_{nm}^a(\mathbf{k})$ ($n \neq m$) is non-Abelian Berry connection. Both of them are geometrical quantities. In other words, there are two geometric forms, the generalized derivatives and Berry connections.

After carefully reading this comment, we think that the referee might have some misconceptions about sum rules. From our perspective, the referee holds the view that a sum rule of the generalized derivative of the dipole matrix element,

$$r_{nm;a}^b = \frac{r_{nm}^a \Delta_{mn}^b + r_{nm}^b \Delta_{mn}^a}{\omega_{nm}} + \frac{i}{\omega_{nm}} \times \sum_l \omega_{lm} r_{nl}^a r_{lm}^b - \omega_{nl} r_{nl}^b r_{lm}^a, \quad (2)$$

will neglect the geometric effects. However, we insist that equation (2) is equivalent to

the equation (1), shown in a generalized derivative form. Next, we will derive equation (2) from equation (1) in detail by using the commutation relation $[r^a, p^b] = i\hbar\delta^{ab}$. We hope this proof process could dispel the misgiving.

We start from the commutation,

$$\begin{aligned}\langle n\mathbf{k}|[r^a, p^b]|m\mathbf{k}'\rangle &= \langle n\mathbf{k}|[r_i^a, p^b]|m\mathbf{k}'\rangle + \langle n\mathbf{k}|[r_e^a, p^b]|m\mathbf{k}'\rangle \\ &= i\hbar\delta^{ab}\delta_{nm}\delta(\mathbf{k} - \mathbf{k}').\end{aligned}\quad (3)$$

When $n \neq m$, the intraband part can be expressed as

$$\begin{aligned}\langle n\mathbf{k}|[r_i^a, p^b]|m\mathbf{k}'\rangle &= i\delta(\mathbf{k} - \mathbf{k}') (p_{nm}^b)_{;k^a} = i\delta(\mathbf{k} - \mathbf{k}') (im\omega_{nm}r_{nm}^b)_{;k^a} \\ &= -m\delta(\mathbf{k} - \mathbf{k}') \omega_{nm}(r_{nm}^b)_{;k^a} - m\delta(\mathbf{k} - \mathbf{k}') r_{nm}^b \Delta_{nm}^a,\end{aligned}\quad (4)$$

which use a relation involving intraband position operator \mathbf{r}_i and simple operator \mathbf{S} ,

$$\langle n\mathbf{k}|[\mathbf{r}_i, \mathbf{S}]|m\mathbf{k}'\rangle = i\delta(\mathbf{k} - \mathbf{k}') (\mathbf{S}_{nm})_{;k}, \quad (\mathbf{S}_{nm})_{;k} \equiv \frac{\partial \mathbf{S}_{nm}}{\partial \mathbf{k}} - i\mathbf{S}_{nm}(\boldsymbol{\xi}_{nn} - \boldsymbol{\xi}_{mm}).$$

The interband part can be expanded by inserting normalized complete basis,

$$\begin{aligned}\langle n\mathbf{k}|[r_e^a, p^b]|m\mathbf{k}'\rangle &= \sum_{l, \mathbf{k}_t} \langle n\mathbf{k}|r_e^a|l\mathbf{k}_t\rangle \langle l\mathbf{k}_t|p^b|m\mathbf{k}'\rangle - \langle n\mathbf{k}|p^b|l\mathbf{k}_t\rangle \langle l\mathbf{k}_t|r_e^a|m\mathbf{k}'\rangle \\ &= \sum_{l, \mathbf{k}_t} (1 - \delta_{nl}) \delta(\mathbf{k} - \mathbf{k}_t) \xi_{nl}^a \cdot p_{lm}^b \delta(\mathbf{k}_t - \mathbf{k}') \\ &\quad - \sum_{l, \mathbf{k}_t} p_{nl}^b \delta(\mathbf{k} - \mathbf{k}_t) \cdot (1 - \delta_{lm}) \delta(\mathbf{k}_t - \mathbf{k}') \xi_{lm}^a \\ &= \sum_{l, l \neq m \neq n} \delta(\mathbf{k} - \mathbf{k}') (\xi_{nl}^a \cdot p_{lm}^b - \xi_{lm}^a \cdot p_{nl}^b) + \delta(\mathbf{k} - \mathbf{k}') (\xi_{nm}^a \cdot p_{mm}^b - \xi_{nm}^a \cdot p_{nn}^b) \\ &= m\delta(\mathbf{k} - \mathbf{k}') (r_{nm}^a \Delta_{mn}^b + i \sum_{l, l \neq m \neq n} (\omega_{lm} r_{nl}^a r_{lm}^b - \omega_{nl} r_{lm}^a r_{nl}^b)).\end{aligned}\quad (5)$$

Here, $\mathbf{p}_{nm} = im\mathbf{r}_{nm}\omega_{nm}$, ω_{nm} is the energy difference of the n^{th} band and m^{th} band.

By using equation (3): $\langle n\mathbf{k}|[r_i^a, p^b]|m\mathbf{k}'\rangle + \langle n\mathbf{k}|[r_e^a, p^b]|m\mathbf{k}'\rangle = 0$ again, we can obtain

$$\begin{aligned}m\delta(\mathbf{k} - \mathbf{k}') \omega_{nm}(r_{nm}^b)_{;k^a} - m\delta(\mathbf{k} - \mathbf{k}') r_{nm}^b \Delta_{mn}^a = \\ m\delta(\mathbf{k} - \mathbf{k}') (r_{nm}^a \Delta_{mn}^b + i \sum_{l, l \neq m \neq n} (\omega_{lm} r_{nl}^a r_{lm}^b - \omega_{nl} r_{lm}^a r_{nl}^b)).\end{aligned}\quad (6)$$

By sorting it, the expansion of the generalized derivative of matrix elements shows as

$$(r_{nm}^b)_{;k^a} = \frac{r_{nm}^a \Delta_{mn}^b + r_{nm}^b \Delta_{mn}^a}{\omega_{nm}} + \frac{i}{\omega_{nm}} \sum_l (\omega_{lm} r_{nl}^a r_{lm}^b - \omega_{nl} r_{lm}^a r_{nl}^b),$$

which is interpreted as a “sum rule”. Here, $\Delta_{mn} = \frac{p_{mm} - p_{nn}}{m} = \frac{\partial \omega_{mn}}{\partial \mathbf{k}}$, represents the difference of group velocity of electrons.

To sum up, we discuss the derivation of formulas and hold the viewpoint that sum rule of the generalized derivative of the dipole matrix element still includes the geometric properties of the Bloch electrons.

Revision: (1) We have added the sentence “*The generalized derivative of the dipole matrix element was expressed as*

$$r_{nm;b}^a(\mathbf{k}) \equiv \frac{\partial r_{nm}^a(\mathbf{k})}{\partial k^b} - i[\xi_{nn}^b(\mathbf{k}) - \xi_{mm}^b(\mathbf{k})]r_{nm}^a(\mathbf{k}).$$

By using the sum rule of generalized derivative of the dipole matrix element, we can rewrite this term as

$$r_{nm;a}^b = \frac{r_{nm}^a \Delta_{mn}^b + r_{nm}^b \Delta_{mn}^a}{\omega_{nm}} + \frac{i}{\omega_{nm}} \times \sum_l \omega_{lm} r_{nl}^a r_{lm}^b - \omega_{nl} r_{nl}^b r_{lm}^a,$$

which is equivalent to the original formula and is more convenient for numerical calculation (Supplementary Note 2).” in Page 19. (2) We have added the above discussions as Note 2 in Supplementary Information. (3) We have revised the manuscript to clearly demonstrate the origin of large SHG or shift-current peaks in different NbOX₂ materials.

Comment 3: “The maximum second-harmonic generation (SHG) susceptibility of NbOCl₂ authors simulated is around 120 pm/V. While the experimental measured value is around 200 pm/V (Fig. 3 (g) of Ref. 19). Authors should reproduce the experimental results before they could explain the origin of large SHG response.”

Response 3: We thank the referee for asking this important question. When we perform the SHG calculations, a small factor η is always introduced in the denominator of the SHG formula (see Method in the main text) to avoid divergence problem. Figure R5 shows the SHG susceptibility calculated using different η . As one can see, the maximum SHG susceptibility of NbOCl₂ gets higher when η gets smaller, and the

experimental results can be reproduced when $\eta = 0.03$. However, the factor η actually implies the lifetime of the carriers, which is hard to be estimated in theory. Therefore, we set $\eta = 0.05$ eV, a value is generally adopted in many other 2D NLO semiconductors [Nano Lett. 17, 5027–5034 (2017); Sci. Adv. 5, eaav9743(2019).], generating the maximum SHG susceptibility of ~ 160 pm/V for NbOCl₂ (Fig. 2f). We note that the η may influence the detailed value, but it will not change the shape of SHG spectrum.

Even though there is a difference of the absolute values between the calculated and the measured SHG, the calculated SHG curve has a similar slope as the experimental measurements, which suggests that the behavior of the calculated SHG as a function of the frequency of incident light is consistent with experimental observations. Therefore, the difference in the absolute values of SHG may not affect the conclusion of the physical mechanism of large SHG in this work.

Fig. R5. The calculated SHG susceptibility $\chi_{yyy}^{(2)}$ as a function of frequency for different factors η .

Revision: we have added the sentences “*In addition, we have compared the available experimental SHG data with our calculated one. Importantly, our calculated spectrum has a similar curvature to the experimental one [19] in spite of the small difference in the absolute values of SHG caused by the approximation used in theoretical calculations. Therefore, the peak B may account for the observed large SHG in NbOCl₂ in experiment.*” in the second paragraph on page 9.

Comment 4: “The authors used a “trick” on the experimental data, as shown in Fig. 2f. They narrowed the range of the experimental data (SHG susceptibility) by 1/4 to make the experimental data fit the calculation well. This is not a scientific way to present raw data.”

Response 4: We thank the referee for this valuable comment. We realize that this data-processing is somewhat misleading. Therefore, we have deleted the experimental data and replotted Fig. 2f.

Comment 5: “Bulk NbOX₂ contains two monolayers in its primary unit (Ref. 22-24), so it is unlikely to displaying ABC stacking along out-of-plane *z* axis.”

Response 5: We thank the referee for raising this problem. Bulk NbOX₂ crystalizes in a monoclinal layered structure, which causes that *a* axis not to be perpendicular to *b* and *c* axis, as shown in Fig. 1a. Therefore, the crystal structure shown in Fig. 1a is not ABC stacking structure.

Revision: We have marked the unitcell of bulk NbOX₂ in Fig. 1a using black-solid lines to eliminate this misunderstanding.

Comment 6: “Authors perform the symmetry derived SHG susceptibility analysis on NbOCl₂ monolayer with orthorhombic C_{2v} point group symmetry. While experimental measurements are performed on NbOCl₂ nanoflakes with multiple layers. Bulk NbOCl₂ adopts C₂ symmetry, such a monoclinic symmetry will also work for NbOCl₂ multiple layers. Compared to orthorhombic C_{2v} symmetry, monoclinic C₂ symmetry without mirror plane has completely different symmetry selection rules. In other words, are SHG simulations performed on C_{2v} NbOCl₂ monolayer comparable to monoclinic NbOCl₂ layers?”

Response 6: We thank the referee for this insightful comment. Firstly, we discuss the symmetry selection rules without the interlayer interactions. Actually, monoclinic C₂ symmetry and orthorhombic C_{2v} symmetry have partially uniform symmetry selection rules due to the double rotational symmetry. To see this point clearly, we plot the crystal structures of bulk NbOCl₂ and monolayer NbOCl₂ in Fig. R6, where the corresponding coordinate systems are displayed. The second-order susceptibility tensor $\chi_{abc}^{(2)}$ (*a*, *b*, *c* represents cartesian coordinates) has 27 components. The spatial symmetry causes $\chi_{abc}^{(2)} = T_{aa'}T_{bb'}T_{cc'}\chi_{a'b'c'}^{(2)}$, where *T* is transformation matrix, and the right side of above

equation satisfies the Einstein's summation rule. For the monoclinic C_2 symmetry, there is a double rotational symmetry about y axis whose transformation matrix can be expressed as

$$T = \begin{pmatrix} -1 & 0 & 0 \\ 0 & 1 & 0 \\ 0 & 0 & -1 \end{pmatrix}.$$

When the left side of above equation walk through all elements, we can list 27 equations. Considering intrinsic permutation symmetry, there is a non-zero component distribution of 3×6 matrix shown in Fig. R6c. On the other hand, for the orthorhombic C_{2v} symmetry, there is an additional mirror symmetry about the yOz plane which satisfies

$$T = \begin{pmatrix} -1 & 0 & 0 \\ 0 & 1 & 0 \\ 0 & 0 & 1 \end{pmatrix},$$

which eliminates three components: xyz , yxz , and zxy , resulting in the 3×6 matrix shown in Fig. R6d. Comparing Figs. R6c and R6d, it can be found that some non-zero components of the second-order susceptibility tensor are all surviving.

Secondly, we discuss the effect of the interlayer interaction. Due to the extremely weak interlayer interaction of NbOCl_2 , stacking multiple layers have similar optical response as monolayer (see Fig. 2f in the main text). Therefore, the SHG simulations performed on monolayer NbOCl_2 are comparable to monoclinic bulk NbOCl_2 .

Fig. R6. **a, b** Crystal structures of bulk and monolayer NbOCl_2 , respectively. Here, x, y, z represents the cartesian axes. Here, C_2 implies 2-fold rotation operation and m is mirror plane. **c, d** The SHG susceptibility tensor $\chi^{(2)}(2\omega; \omega, \omega)$ of monoclinic C_2 point group and orthorhombic C_{2v} point group.

Revision: (1) We have added the following sentence “*Due to the symmetry limitation for the second-order susceptibility tensor and the weak interlayer coupling, the non-zero tensor components which we are interested in the bulk NbOX₂ are almost same in the monolayer NbOX₂ (Supplementary Fig.3 and Supplementary Note 1).*” in Page 6. (2) We have added above symmetry discussions as the Note 1 and Fig. R6 as Fig. 3 in Supplementary Information.

Comment 7: “In page 8, ‘SHG strength of NbOCl₂ is almost independent of the sample thickness.’ In Fig. 3e of Ref. 19, the significant thickness dependent SHG intensity is observed in NbOCl₂ nanoflakes. What is the different between ‘SHG strength’ in current work and ‘SHG intensity’ from Ref. 19?”

Response 7: There are two differences between the ‘SHG strength’ in our work and the ‘SHG intensity’ in Ref. 19.

First, the SHG response observed in experiment is a total effect contributed by each vdW layer of a sample. On one hand, for bulk NbOCl₂ or NbOCl₂ multilayer films, the interlayer stacking cannot bring new inversion symmetry. Therefore, each vdW layer can have a positive contribution to the total SHG response, leading to the thickness dependent SHG intensity reported in Ref. 19. However, due to the accumulation effect, it is not very meaningful to directly compare the SHG response of the samples with different thickness (at least in theory). In our work, we define the effective SHG strength as the averaged value of the SHG intensity at each vdW layer (see the Method part of the main text) and use the SHG strength to compare the SHG response of different samples. Because of the weak interlayer coupling, the SHG strength is almost independent of the sample thickness.

Second, the SHG intensity experimentally measured is characterized by the intensity of the outgoing light. In a nutshell, $I \propto |P_{NL}(2\omega)|^2 \propto |N\chi_{monolayer}^{(2)}|^2$. Here, I is the outgoing light intensity, $P_{NL}(2\omega)$ is the intensity of polarization, N is the number of layers, and $\chi_{monolayer}^{(2)}$ is the SHG susceptibility of monolayer structure. Therefore, the SHG intensity establishes the square relationship with the number of layers, as shown in Fig. 3e of Ref. 19. Differently, the SHG strength in our work is the second-order susceptibility

instead of the intensity of the outgoing light.

Revision: We have added the following statement: “*In experiment, the SHG intensity is characterized by the intensity of the outgoing light (I). The relationship between the SHG intensity measured in experiments and the SHG strength discussed in this paper can be expressed as $I \propto |P_{NL}(2\omega)|^2 \propto |N\chi_{monolayer}^{(2)}|^2$,” in page 20.*

Comment 8: “In Page 10, authors mentioned ‘NbOCl₂ a promising material in the field of energy conversion’. In fact, ferroelectric photovoltaic effect of NbOI₂ nanoflakes has been measured by experiment (Appl. Phys. Lett. 119, 033103 (2021)). Even NbOI₂ has a much smaller band gap, it still displays a quite weak photocurrent. Compared to NbOI₂, NbOCl₂ has larger band gap and therefore is more insulating. Is insulating NbOCl₂ really promising for solar energy conversion?”

Response 8: We thank the referee for raising this important question, which is actually a key question in the field of nonlinear optics.

Firstly, in the reference [Appl. Phys. Lett. 119, 033103 (2021)] raised by referee, the transport properties of the NbOI₂ device under illumination have been systematically studied. It is found that the NbOI₂ device can establish optically active reconfigurable room temperature rectification. Different from this work, we investigate the shift current responses of NbOX₂, where the current is created by the changes in the position of the electron wave-packet during light excitation process instead of applying voltage.

Secondly, we want to emphasize that there are many factors which can influence the solar energy conversion, such as the physical mechanism of photocurrent, the optical absorption of material, the concentration of defects, the transport property of material and the contact property of the device. Therefore, one particular property of the material cannot determine the overall efficiency of a photovoltaic device. Improving efficiency requires a fine balance of all these factors that affect the energy conversion efficiency of the whole device. From this point of view, despite that the measured photocurrent of the NbOI₂ device is relatively weak [Appl. Phys. Lett. 119, 033103 (2021)] at the present stage, it is still difficult to assert that the energy conversion efficiency of the

NbOX₂ devices is too low to be useful after reasonable consideration of other factors affecting the efficiency.

Finally, we fully agree with the referee that it is too early to claim NbOCl₂ as a promising material in the field of energy conversion, especially without future experimental efforts. On the other hand, the NbOCl₂ may be useful to work as photo-detector or sensor at specific photon-energy regions.

Revision: (1) We have modified the sentence to “*Compare to other 2D material like TMDs where NLO responses are suppressed in even-layer thickness, NbOCl₂ maybe have some advantages in realistic energy applications own to the layer-independent σ .*” in Page. 10. (2) We found that the reference [Appl. Phys. Lett. 119, 033103 (2021)] mentioned by the referee is closely related to our work, we have added the discussions “*Undoubtedly, there are many other effects should be under consideration in realistic applications, like reported rectification in vdW NbOI₂ sheets [43]*” in page 10 and added this reference as Ref. 43 in the main text.

Comment 9: “Based on my knowledge of X-ray diffraction measurement, I did not see any FE-to-AFE phase or symmetry change induced diffraction peak change. How can authors conclude that there is a FE-to-AFE phase transition in NbOCl₂ single crystal sample based on X-ray diffraction measurement?”

Response 9: We appreciate the referee for raising this critical question, which is also one of the key findings of our study. The powder X-ray diffraction is a general method of X-ray diffraction measurement for the structural determination used in experiments. As shown in Fig. R7, the differences of powder X-ray diffraction pattern between the FE and AFE phases are subtle, therefore, both powder X-ray diffraction and Raman testing are unable to effectively distinguish between them, therefore, they cannot explain the large phase transition indicated by the SHG measurement. For example, the noncentrosymmetric space group *C2* and the centrosymmetric space group *C2/c* belong to the same diffraction group. The positions of their diffraction peaks are consistent. *Therefore, one of the best ways to accurately determine the structures of these two phases is the single crystal X-ray diffraction [e.g., see Nature Chemistry, 15, 641*

(2023)]. Our experimental team has solid experience for single crystal X-ray diffraction. The typical steps in single crystal analysis involve the calculation of the lattice parameters from the positions of reflections and analyzing the diffraction intensities at each Bragg position. By applying extinction rules, we can select the space groups and ultimately determine the accurate crystal structure, which is also validated by the other experimental techniques, such as the observation of SHG signals, as presented in the present study. *We have put all the structural details in the supplementary Tables S1-S10 for other researchers to use or repeat our results.*

Supplementary Fig. 6 displays the X-ray diffraction patterns of the single crystal NbOCl_2 in the $0kl$ direction at 500 K and 5.7 GPa. This figure serves two main purposes. Firstly, it demonstrates a clear, regular, and without noticeable tailing diffraction pattern, indicating a high-quality single crystal (The diffraction rings observed in Supplementary Fig. 6 are caused by the presence of the gasket and the pressure-transmitting medium). Secondly, it shows the Bragg diffraction pattern (021), which indicates that the c -axis of the unit cell expands to twice its original length, as shown in Fig. 1b in the main text. In the FE phase of NbOCl_2 , only two Nb-O-Cl octahedra are allowed along the c -axis. After the phase transition, the AFE phase of NbOCl_2 allows the presence of four Nb-O-Cl octahedra. This is the reason for the AFE phase having a polarization arrangement of $\uparrow\uparrow\downarrow\downarrow$ instead of $\uparrow\downarrow$. Our total energy calculations also confirm the $\uparrow\uparrow\downarrow\downarrow$ is energetic more favorable than $\uparrow\downarrow$ and other configurations (see Table R1).

Fig. R7. The collected powder x-ray diffraction patterns (*left*) and the vibrational Raman modes

(right) as function of pressure of NbOCl₂

Revision: We have added the above discussion in the page 12 and legend of Supplementary Fig. 6 in the revised manuscript.

Comment 10: “Measurement of NbOCl₂ crystal under pressure up to 10 GPa, has been conducted in Fig. S20 of Ref. 19. NbOCl₂ is known for its significant optical nonlinearity and large SHG response. In current work, authors reported SHG responses of NbOCl₂ crystal can be completely suppressed under the pressure around 5.7 GPa. So, from high SHG response to zero SHG response, is pressure really an effective way to ‘fine tune’ nonlinear optical responses of NbOCl₂?”

Response 10: We really appreciate the referee for this question, which is related to the above comment. In previous study of Ref.19, NbOCl₂ did exhibit large SHG response *without* pressure. Also, in the previous study, the Raman spectra at the pressure up to 10 GPa indeed showed no significant changes, and the new collections of powder x-ray diffraction pattern of NbOCl₂ under pressure (Fig. R7) also support that there seem no phase changes below 10 GPa. However, as explained in Response 9, single-crystal X-ray diffraction is very important to accurately identify the small bond-change-induced phase transition. Also, the results of SHG measurements in our present study show the clear changes of SHG response under pressure, strongly invalidating the conclusion from Raman spectra in Ref. 19, which stimulate us to use more advanced technique (e.g., single-crystal analysis) to explore the reason of SHG changes. Eventually, our study demonstrates that pressure can be an effective way to tune the nonlinear optical response as a result of FE-to-AFE phase transition in NbOCl₂. We also think this is one of our major novelties beyond previous study published in Nature.

Revision: (1) We have added the sentence “*We emphasize that the subtle structure changes cannot be well detected by powder X-ray diffraction patterns (Supplementary Fig. 7).*” in page 12. (2) We have added Fig. R7 (left panel) as Supplementary Fig. 7.

Comment 11: “Antiferroelectric (AFE with $\uparrow \uparrow \downarrow \downarrow$ polarization arrangement) phase reported in current work is more like two short-periodic FE single domain with

opposite polarization, separated by 180 FE domain wall (Fig. S11 of Ref. 18). While each single FE domain should still have non-zero and detectable SHG signal. Based on authors logic, $\uparrow\uparrow\downarrow\downarrow$ AFE phase is more stable than $\uparrow\downarrow$ AFE phase, will $\uparrow\uparrow\downarrow\downarrow\downarrow\downarrow$ or $\uparrow\uparrow\uparrow\uparrow\downarrow\downarrow\downarrow\downarrow$ AFE phase also exist in NbOCl_2 crystal under pressure?”

Response 11: We thank the referee for raising this question. First of all, we want to clarify that the $\uparrow\uparrow\downarrow\downarrow$ configuration of AFE phase is *determined by the single-crystal X-ray diffraction analysis* (NOT from our calculations). At present, we have not found other AFE phases in our experiment. We agree with the referee that their might have local SHG signal at each local inversion symmetry-breaking domain, forming a hidden NLO effect. But the overall SHG intensity of the whole AFE phase should be zero, as also confirmed in our experimental measurements. This kind of hidden second-order NLO effects have been discussed in the revision (page 14 and revised Fig. 4), according to the referee’s comment.

Following the referee’s suggestion, we also have calculated the total energy of the AFE phases with four different configurations (AFE1: $\uparrow\downarrow$, AFE2: $\uparrow\uparrow\downarrow\downarrow$, AFE3: $\uparrow\uparrow\downarrow\downarrow\downarrow\downarrow$, AFE4: $\uparrow\uparrow\uparrow\downarrow\downarrow\downarrow\downarrow$) at 6 GPa, and the results are listed in Table R1. As one can see, the $\uparrow\uparrow\downarrow\downarrow$ AFE phase has the lowest total energy, which suggests that the $\uparrow\uparrow\downarrow\downarrow$ AFE phase is the most stable phase.

Table R1. The total energy of the bulk NbOCl_2 (in meV/unit cell) with five different configurations under 6 GPa pressure. Here, the energy of AFE2 is set as zero reference level, and the energies are compared with unified lattice size.

Configurations	Relative Energy
$\uparrow\uparrow\uparrow\uparrow$	18.8
$\uparrow\downarrow\uparrow\downarrow$	62.8
$\uparrow\uparrow\downarrow\downarrow$	0
$\uparrow\uparrow\uparrow\downarrow\downarrow\downarrow$	39.2
$\uparrow\uparrow\uparrow\uparrow\downarrow\downarrow\downarrow\downarrow$	36.1

Revision: (1) We have added the sentences “*Our calculations also reveal that this AFE*

state is more stable than other metastable AFE states [45]” in page 12. (2) We have a detailed discussion on the possible hidden second-order NLO responses in page 14, along with the revised the Figure 4.

Response for the third referee:

Comment 1: “The manuscript theoretically investigates the band structures of niobium oxide dihalides and their optical properties such as SHG coefficients, absorption strength, shift current coefficients. The calculation gives insight of these optical properties such as the origin of the optical features. In addition, the authors claimed to experimentally observe the antiferroelectric (AFE) phase and demonstrate how to tune from ferroelectric (FE) phase to AFE phase. Overall, the calculation and explanation of the optical features are insightful. However, the most interesting part to Nature Communications, the claim of AFE phase, is lack of solid evidence in this manuscript. A few issues should be addressed for reconsideration for publication.”

Response 1: We thank the referee for the careful review on our manuscript and the positive comments of the quality of our manuscript. In the following, we will try our best to explain the issues raised by the referee.

Comment 2: “X-ray diffraction should be direct evidence to reflect the structures of AFE and FE phases. In this work, the authors measured the X-ray diffraction at 500 K and at 5.7 GPa, respectively. However, only one figure S5 in the supplementary information to show the pattern does not justify differentiating the subtle change between AFE and FE. I suggest the authors show the theta-scan figures and the fitting results of the key diffraction peaks. For example, by Comparing with 300 K and 500K (or with atmosphere pressure and 5.7 GPa), could the authors clearly identify the difference between the key peaks corresponding to the AFE and FE phase? A comparison between the AFE and FE diffraction data will help to clarify if the subtle change is in the error range for fitting.”

Response 2: We appreciate the referee for this important question. Generally, the

powder X-ray diffraction patterns are an effective way to determine the phase transitions, in particular, for the reconstructive phase transitions with significant diffusion of atoms in materials. Here, for van der Waals layered crystals of NbOCl₂, our observations support that the phase transition is to be displacive under pressure with subtle change of atoms, which cannot be well seen from the collected in situ high pressure powder X-ray diffraction and Raman spectra (see Figure R7). Accordingly, it is difficult to understand the large SHG changes just using powder X-ray diffraction and Raman spectra. *Therefore, one of the best ways to accurately determine the structures of these two phases is the single-crystal X-ray diffraction [e.g., see Nature Chemistry, 15, 641 (2023)].* Our experimental team has solid experience for single crystal X-ray diffraction. The typical steps in single crystal analysis involve the calculation of the lattice parameters from the positions of reflections and analyzing the diffraction intensities at each Bragg position. By applying extinction rules, we can select the space groups and ultimately determine the accurate crystal structure, which is also validated by the other experimental techniques, such as the observation of SHG signals, as presented in the present study. *We have put all the structural details in the supplementary Tables S1-S10 for other researchers to use or repeat our results.*

Figure S5 displays the X-ray diffraction patterns of the single-crystal NbOCl₂ in the 0kl direction at 500K and 5.7 GPa. This figure serves two main purposes. Firstly, it demonstrates a clear, regular, and without noticeable tailing diffraction pattern, indicating a high-quality single crystal (The diffraction rings observed in Fig. S5 are caused by the presence of the gasket and the pressure-transmitting medium). Secondly, it shows the Bragg diffraction pattern (021), which indicates that the *c*-axis of the unit cell expands to twice its original length, as shown in Fig. 1b. In the FE phase of NbOCl₂, only two Nb-O-Cl octahedra are allowed along the *c*-axis. After the phase transition, the AFE phase of NbOCl₂ allows the presence of four Nb-O-Cl octahedra. This is the reason for the AFE phase having a polarization arrangement of $\uparrow\uparrow\downarrow\downarrow$ instead of $\uparrow\downarrow$. Our total energy calculations also confirm the $\uparrow\uparrow\downarrow\downarrow$ is energetic more favorable than $\uparrow\downarrow$ and other configurations (see Table R1 above).

Fig. R7. The collected powder x-ray diffraction patterns (*left*) and the vibrational Raman modes (*right*) of as function of pressure of NbOCl₂.

Revision: We have added the above discussion in the page 12 and legend of Supplementary Fig. 6 in the revised manuscript.

Comment 3: “The authors measured the second harmonic generation (SHG) of NbOCl₂ as a function of pressure and temperature. While the crystal was under high pressure or temperature, the SHG efficiently dramatically dropped. The authors attributed the diminishing SHG to AFE with inversion symmetry. Other possible scenarios should be ruled out to support this scenario as indirect evidence. For example, in Fig. 2f, the authors show the SHG coefficient spectra, which exhibit an edge around 2 eV. SHG intensity is sensitive for wavelengths around the edge. Will the SHG intensity vary due to the shift of the edge (or shift of the bandgap corresponding to B peak) for different temperature or pressure? In other words, is there dependence of temperature or pressure on the SHG coefficient spectra?”

Response 3: We thank the referee for raising this nice question. First, *it is worth emphasizing that the FE-to-AFE phase transition is directly observed by the single-crystal X-ray diffraction analysis technique (not from the calculations), as demonstrated in Response 2.* Single-crystal X-ray diffraction analysis technique is one of the most powerful and accurate methods to determine the fine structures of crystals [e.g., see Nature Chemistry, 15, 641 (2023)].

Second, in order to discuss the influence of pressure, we have performed calculations on the bulk NbOCl_2 under 0 GPa, 2 GPa, 4 GPa, and 6 GPa. It is found that the lattice constants a , b , and c shrink by 9.1%, 1.6% and 1.8% respectively when pressure is applied from 0 GPa to 6 GPa. Here, a is lattice constant along van der Waals layers. Figure R8 shows the calculated total density of states (TDOS) of the bulk NbOCl_2 under different pressures. As one can see, there are only small changes of the TDOS of the bulk NbOCl_2 under different pressures except for decreasing bandgap under pressure, implying the similarity of the band structures. Therefore, the B peak of SHG spectra will move left a little, ~ 0.05 eV, when pressure increases every 2 GPa from the aspect of numerical simulation, which means the SHG will change continuously or regularly. However, in our experimental SHG measurement, the response keeps an average level under low pressure and diminish dramatically to zero over a certain pressure as shown in Fig. 4d in the main manuscript. *Hence, the speculation, dependence of pressure on SHG spectra makes the SHG diminish dramatically, could be ruled out.*

Finally, we consider the influence of the thermal expansion. After checking the lattice constants of crystal NbOCl_2 under 100, 300, 500K, we find the thermal expansion influence slightly. Therefore, the band structure of the bulk NbOCl_2 almost has unchanged when considering the thermal expansion, and the SHG spectra should change slightly correspondingly. However, in our experiments, the observed SHG hold the line when the temperature increase from 300 K to 400 K and dramatically jump to zero as temperature reach 500 K (see Supplementary Fig. 8). By comparison, *the possibility of temperature shifts the edge of SHG to zero could be ruled out.*

Fig. R8. The TDOS of the bulk NbOCl_2 under different pressures (*left panel*) and temperatures (*right panel*).

Revision: We have added the following discussions “*In addition, to support that FE-to-AFE phase transition causes dramatical descent of SHG, the other possible scenarios, variation of band edge under external fields, have be ruled out (Supplementary Fig. 9)*” in page 13 and added Fig. R8 as Fig. 9 in supplementary information.

Comment 4: “Although the authors calculated that the total energy of AFE is only 2 meV/u.c higher than that of FE as shown in Fig. 4f, it is not straightforward to me that over a certain temperature or pressure, the whole crystals would be completely transformed into a slightly higher energy AFE state, rather than co-existence of AFE and FE states in random order. Although temperature and pressure provide a driving force to overcome the barrier energy, they do not specifically drive the system toward AFE state. Typically for ferroelectric or ferromagnetic materials, the ferroic property disappears above a transition temperature (Curie temperature). Could it be possible that NbOCl₂ simply becomes centrosymmetric PE phase without local spontaneous polarization above a certain temperature or pressure?”

Response 4: We agree with the referee that the tiny difference in total energy between FE and AFE phases and the small energy barrier cannot prove that the external fields can lead to the FE-to-AFE phase transition. Again, we would like to emphasize that *the FE-to-AFE phase transition is identified by combining the SHG measurements and the single-crystal X-ray diffraction analysis technique instead of the theoretical calculations.*

The change of the normalized SHG intensity with increasing pressure or temperature shows three different stages, as shown in Figs. 4d and 4e in the main text. When the pressure is between ~3 and ~5.7 GPa or the temperature is between ~400 and 450 K, the system stays at stage II. As the referee pointed out, there is indeed the states with the co-existence of AFE and FE states in random order at this time, as reflected by the normalized SHG intensity between 1 and 0.

When the pressure exceeds 5.7 GPa or the temperature exceeds 450 K, the normalized SHG intensity decreases to zero, which implies that the system transforms into a state

with the centrosymmetric crystal structure. As the referee pointed out, the FE property usually disappears when the external fields exceed a critical value, which also can cause the zero SHG intensity. However, in our work, the single-crystal X-ray diffraction analysis identifies that the crystal structure is a AFE phase (at least within our used pressure region) rather than the paraelectric (PE) phase without local spontaneous polarization as in the usually case. This FE-to-AFE phase transition is one of the highlight points of this work. We agree with the referee that under much higher pressure or temperature, the AFE might be eventually converted to the PE phase. However, in our current study, we focus on the relatively small pressure or temperature region, which is easier to realize for device application.

Finally, we note that, in the revision, we have added new experimental test on the NbOI₂ system. Differing from the NbOCl₂, only FE-to-PE can be detected in NbOI₂ under pressure (see **Response 6** to referee 1 or revised Figure 4 in the main text), indicating that the FE-to-AFE transition is unique in NbOCl₂.

Revision: We have added the sentence “*the collected single-crystal X-ray diffraction reveals that NbOCl₂ undergoes a novel FE-to-AFE phase transition rather than simply to the centrosymmetric without local spontaneous polarization (Supplementary Fig. 6 for detailed single-crystal X-ray diffraction patterns)*” in page 12.

Comment 5: “For the band structure calculation, the authors mentioned a scissor operation is used. The authors should show how much energy is shifted according to the measured bandgap from which literature. Or it would be better that the authors experimentally determine the bandgap of their crystals. For example, in Fig. 2f, the experimental data from Ref. 19 was blue-shifted 0.6 eV for comparison. Does the bandgap already shift to the right energy referred to Ref. 19 while a 0.6 eV shift is still necessary for comparing SHG coefficient spectra?”

Response 5: We thank the referee for raising this question. Comparing to optical bandgap by experiment, PBE functional will underestimate the bandgap while HSE06 functional is too expensive for the NLO calculations. Therefore, the scissor operations were employed to modify the bandgap artificially where the mechanism remains the

same except response values. Considering the systematical simulation on NbOX₂ (X = Cl, Br, and I), we have used a scissor operation from HSE06 functional calculations, rather than the measured bandgap from other literature, which have been stated in Method part. Also, we realize that blue-shifting experimental data is a little misleading and isn't a scientific way to show raw data.

Revision: We have deleted the experimental data and replot the Fig. 2f.

Comment 6: “In the introduction ‘Among numerous materials, compounds without inversion symmetry can exhibit various nonlinear optical (NLO) responses.’ might require revision. NLO responses also include third order nonlinearity, which exists all kinds of materials. The authors might mean second-order nonlinear responses rather than NLO responses.”

Response 6: We thank the referee for raising this problem. We have changed this sentence to “*Among numerous materials, compounds without inversion symmetry can exhibit various second-order nonlinear optical (NLO) responses.*”.

Comment 7: “In page 7 ‘Again, this strong anisotropic feature is absent in many existing 2D materials.’ might be wrong. Most 2D material crystals exhibit anisotropic features if they are non-centrosymmetric. The angle-resolved SHG pattern reflects the group symmetry of the crystal. Graphene and Bi₂Se₃ in Ref. 20, 21 exhibit inversion symmetry. The SHG originates from the surface or interface where the symmetry breaks.”

Response 7: We agree with the referee. We have deleted this sentence in the revised manuscript. We thank the referee for pointing out the problem.

In conclusion, we wish the referees can be satisfied with our response and recommend the acceptance our revised manuscript in Nature Communications.

Revision summary

1. We have made a major revision in the manuscript (marked as blue), along with the updated Figures 1, 2, and 4.
2. We have added Supplementary Figs. S7-S9, S12-S13, and S15 as new figures, Note 1-2 and Tables S6-S10 in the Supplementary Information.

REVIEWER COMMENTS

Reviewer #1 (Remarks to the Author):

After reading the reply letter and the revised manuscript, it seems to me that the authors have addressed most of the comments of the reviewers. therefore, I suggest acceptance.

Reviewer #2 (Remarks to the Author):

In the revised manuscript, the authors have conducted additional experimental measurement and theoretical simulations to address reviews' comments and suggestions. The overall revision work is solid and satisfactory. The current manuscript is in good published form, after authors can further address following minor issues:

1. In the abstract section, "we achieve a reversible ferroelectric-to-antiferroelectric (paraelectric) phase transition in NbOCl₂ (NbOI₂) under a certain region of external pressure" is not a solid statement. NbOI₂ undergoes ferroelectric-to-paraelectric phase transition under pressure.
2. In line 144-146, "... NbOX₂ in FE phase (left panel), which may be converted to a possible AFE phase or a PE (right panel) phase via applying external fields". As far as I know, electric field can only drive structural transition from AFE or PE phase to lower-symmetry FE phase. Transition from FE to AFE or PE phase by electric field, is less likely.
3. I agree with authors that the most outstanding finding in current work is "discovering the FE-to-AFE phase transition driven by moderate pressure (5.7 GPa) or temperature (450 K)". Especially AFE phase in NbOCl₂ with $\uparrow\uparrow\downarrow\downarrow$ polarization arrangement, has never been reported before. Structural characterizations of AFE phase in NbOCl₂ under high pressure or high temperature has been conducted using single crystal X-ray diffraction method. Since AFE phase also appears in NbOCl₂ sample around 500 K at ambient condition, high-resolution STEM can be performed to determine the polar displacement directions of Nb atoms in AFE-NbOCl₂ phase (Fig. 1 of Ref. 19), which can provide more convincing and visualizable evidence for presence of such a rare AFE phase in NbOCl₂.

Reviewer #3 (Remarks to the Author):

The authors addressed most of my comments. However, the most critical one (comment 2 of reviewer 3) was not addressed satisfactorily. The authors tried to convince the reviewers they have good team to perform single-crystal X-ray diffraction to see the subtle changes between FE and AFE. They emphasized that the fitting parameters in Table 1- Table 10 in the supplementary file could be used for other researchers for repeating the results. However, they did not show the fitting results on the raw data such as Fig. S6. Not to mention that they did not show the X-ray diffraction patterns of single crystal NbOI₂.

My original comment suggested the authors showing the Bragg diffraction pattern when NbOCl₂ is in FE state at ambient condition for comparison. It will be helpful to convince the readers that this technique could identify the subtle changes. For example, the authors mentioned the Bragg diffraction pattern (021), which indicates that the c-axis of the unit cell expands to twice its original length, as shown in Fig. 1b in the main text. Does it mean (021)

should disappear in FE state at room temperature and ambient pressure? If this is the case, that will be convincing to show the Bragg diffraction pattern of the crystal in FE state.

I do not recommend publication in Nature Communication until the results of single-crystal X-ray diffraction are presented convincingly.

Response Letter

We thank the referees for their careful review on the revised manuscript. We have carefully considered all the comments and made responses as follows. Meanwhile, the changes in the manuscript are marked as blue color.

Response for the first referee:

Comment 1: “After reading the reply letter and the revised manuscript, it seems to me that the authors have addressed most of the comments of the reviewers. therefore, I suggest acceptance.”

Response 1: We thank the referee for recommending this manuscript to be published on *Nat. Commun.* Many thanks for their professional review on the manuscript.

Response for the second referee:

Comment 1: “In the revised manuscript, the authors have conducted additional experimental measurement and theoretical simulations to address reviews’ comments and suggestions. The overall revision work is solid and satisfactory. The current manuscript is in good published form, after authors can further address following minor issues.”

Response 1: We thank the referee for the positive comments and the recommendation of our work. In the following, we have properly addressed the issues mentioned by the referee one by one.

Comment 2: “In the abstract section, ‘we achieve a reversible ferroelectric-to-antiferroelectric (paraelectric) phase transition in NbOCl_2 (NbOI_2) under a certain region of external pressure’ is not a solid statement. NbOI_2 undergoes ferroelectric-to-paraelectric phase transition under pressure.”

Response 2: We have changed this sentence to “*we achieve a reversible ferroelectric-to-antiferroelectric phase transition in NbOCl₂ and a reversible ferroelectric-to-paraelectric phase transition in NbOI₂ under a certain region of external pressure.*” in Page 2.

Comment 3: “In line 144-146, ‘... NbOX₂ in FE phase (left panel), which may be converted to a possible AFE phase or a PE (right panel) phase via applying external fields.’ As far as I know, electric field can only drive structural transition from AFE or PE phase to lower-symmetry FE phase. Transition from FE to AFE or PE phase by electric field, is less likely.”

Response 3: We agree with the referee. We have changed this sentence to “*which may be converted to a possible AFE or PE phase (right panel) via applying external approaches, like pressure.*” in Page 6.

Comment 4: “I agree with authors that the most outstanding finding in current work is ‘discovering the FE-to-AFE phase transition driven by moderate pressure (5.7 GPa) or temperature (450 K)’. Especially AFE phase in NbOCl₂ with ↑↑↓↓ polarization arrangement, has never been reported before. Structural characterizations of AFE phase in NbOCl₂ under high pressure or high temperature has been conducted using single crystal X-ray diffraction method. Since AFE phase also appears in NbOCl₂ sample around 500 K at ambient condition, high-resolution STEM can be performed to determine the polar displacement directions of Nb atoms in AFE-NbOCl₂ phase (Fig. 1 of Ref. 19), which can provide more convincing and visualizable evidence for presence of such a rare AFE phase in NbOCl₂.”

Response 4: We thank the referee for this very nice suggestion. Indeed, it will be more intuitively to provide the STEM images of the AFE-NbOCl₂. Unfortunately, it is quite challenge for us to perform the high-resolution STEM under high temperature. Actually, the high-quality STEM images of NbOCl₂ at ambient condition shown in *Nature* 613, 53-59 (2023) are done by one of the coauthors of our manuscript after many hard attempts. In addition to the highly nontrivial experimental technics for high-

temperature STEM, the radiation resistance of NbOCl₂ is not good and the bombardment of electrons can break its crystal structure, which make this task even harder under a high temperature. Moreover, given that the STEM can only present the local atomic structure information, the global symmetry and accurate bond information can be better provided by the X-ray single-crystal diffraction, as we have done in the present study.

We apologize that we cannot provide the high-temperature STEM images of the AFE-NbOCl₂. However, we believe that the crystal structures have been well determined by combining the results of X-ray single crystal diffraction (Fig. S6-S7, Table S1-S21), second-harmonic generation (SHG) spectrum (Fig. 4d and Fig. 4e), and density functional calculations.

Based on the Referee's suggestion, we have added some of the above discussion in the Page 11 and Page 17 in the revision.

Response for the third referee:

Comment 1: “The authors addressed most of my comments. However, the most critical one (comment 2 of reviewer 3) was not addressed satisfactorily. The authors tried to convince the reviewers they have good team to perform single-crystal X-ray diffraction to see the subtle changes between FE and AFE. They emphasized that the fitting parameters in Table 1- Table 10 in the supplementary file could be used for other researchers for repeating the results. However, they did not show the fitting results on the raw data such as Fig. S6. Not to mention that they did not show the X-ray diffraction patterns of single crystal NbOI₂.

My original comment suggested the authors showing the Bragg diffraction pattern when NbOCl₂ is in FE state at ambient condition for comparison. It will be helpful to convince the readers that this technique could identify the subtle changes. For example, the authors mentioned the Bragg diffraction pattern (021), which indicates that the c-axis of the unit cell expands to twice its original length, as shown in Fig. 1b in the main

text. Does it mean (021) should disappear in FE state at room temperature and ambient pressure? If this is the case, that will be convincing to show the Bragg diffraction pattern of the crystal in FE state.

I do not recommend publication in Nature Communication until the results of single-crystal X-ray diffraction are presented convincingly.”

Response 1: We apologize for the imperfect response of the comment about structural characterization. Now we understand that the referee wants to see all the single-crystal X-ray diffractions before and after phase transition.

Following referee’s suggestion, we show the collected $(0, k, l)$ reflections of NbOCl_2 in **Fig. R1**, with some reflections marked by their corresponding indices. Under high temperature or pressure (**Fig. R1b** and **R1c**), additional reflections along the $(0, \pm 2, l)$ lines appear (**Fig. R1b** and **R1c**), *i.e.*, the miller indices $(0, k, l)$ regarded to the structure at ambient condition is transformed to $(0, k, 2l)$ under high temperature or pressure. Thus, the unit cell will be doubled along c axis when increasing temperature or applying pressure, which is consistent with the unique AFE phase reported here because the $\uparrow\downarrow\downarrow$ polarization arrangement just requires the doubled unit cell along c axis of the FE phase.

Figure R1. The $(0, k, l)$ reflections of NbOCl_2 obtained by single crystal X-ray diffraction. (a) at ambient condition; (b) at 500K; (c) at 5.7 GPa.

The detailed structural information can be obtained by fitting results on the original diffraction pattern (see Method section in the main text). At the ambient condition, for better understanding the structural information, we list two possible unit cells with different numbers of formulas per unit cell (Z value), and the detailed fitting results are

shown in **Table R1**. We can see that NbOCl_2 prefers the structure with $Z=4$ instead of $Z=8$ ($Z=4$: $R=0.034$, $wR=0.077$; $Z=8$: $R=0.057$, $wR=0.109$) after performing the structural refinements, with weak reflections caused by structural distortion, which is consistent with the structure reported in previous literature [e.g., see *Angewandte Chemie* 1964, 76, 833-849; *Nature* 2023, 613, 53-59]. Under high temperature or high pressure, however, the extra reflections suggest the doubled Z value of the crystal structures, implying the structural phase transitions of NbOCl_2 . As shown in **Table R2**, the fitting results after performing the structural refinements show that the structures under high temperature (500 K, $R=0.078$, $wR=0.160$) or pressure (5.7 GPa, $R=0.089$, $wR=0.227$) prefer $Z=8$.

Figure R2. The $(0, k, l)$ reflections of NbOI_2 obtained by single crystal X-ray diffraction. (a) at ambient condition; (b) at 10.7 GPa.

Besides the case of NbOCl_2 , we also show the original single X-ray diffraction pattern of NbOI_2 in **Fig. R2**. As we can see, NbOCl_2 and NbOI_2 share the same symmetry at ambient condition, but some weak reflections, e.g. $(0, \pm 2, \pm 1)$, are barely seen in **Fig. R2**. Again, we emphasize that the complete structural phase transition of NbOI_2 is difficult to be obtained by comparing the two diffraction patterns directly. Some important structural information caused by the inconspicuous changes in diffraction patterns, e.g., the changes of the reflection intensity of $(0, 0, 3)$, can only be refined by fitting results on the original diffraction pattern (see Method section in main text). The fitting results under 10.7 GPa support a change from a non-centrosymmetric structure with $C2$ space group to a centrosymmetric structure with $C2/m$ space group, which is also consistent with the SHG measurements (see **Fig. 4e** in main text). **Tables R3-R8** and **Tables S12-S16** (see Supplemental Information) lists the detailed structural information of NbOI_2 at the ambient condition and under 10.7 GPa after performing the structural refinements,

respectively, which clearly shows the detailed structural information of the FE phase for NbOI₂ at the ambient condition and the PE phase for NbOI₂ under pressure.

As a response, we have added **Figs. R1, R2** as **Figs. S6, S7** and **Tables R1, R4-R8** as **Table S1, Table S7-S11** (we note that the structure information of PE-phase NbOI₂ was already provided as **Tables S12-S16** in the supplementary information). Besides, we also have added the above discussions to caption of Fig. S6 and S7 in Supplementary Information.

We wish these additional data and discussions will eliminate any concerns and hesitation of the referee when recommending the publication of our work.

Revision summary

1. We have made a minor revision in the manuscript (marked as blue).
2. We have updated Supplementary Fig. S6 and added Supplementary Fig. S7 as a new figure, and added Tables S1 and S7-S11 as new tables in the Supplementary Information.

Table R1. Experimental crystallographic data for NbOCl₂ at ambient conditions obtained by single-crystal X-ray diffraction.

	NbOCl ₂ , Z=4	NbOCl ₂ , Z=8
Chemical formula	NbOCl ₂	NbOCl ₂
M_r	179.81	179.81
Crystal system, space group	Monoclinic, C2	Monoclinic, C2
a, b, c (Å)	12.8412(15), 3.9025(4), 6.7148(9)	12.8452(17), 3.9028(4), 13.4311(19)
α, β, γ (°)	90, 105.830(5), 90	90, 105.846(5), 90
V (Å ³)	323.74(7)	647.74(14)
Z	4	8
Density (Mg/m ³)	3.689	3.688
Wavelength (Å)	0.71073	0.71073
μ (mm ⁻¹)	5.089	5.087
Absorption correction	Multi-scan	Multi-scan
T_{\min}, T_{\max}	0.62, 0.85	0.60, 0.85
No. of measured, independent and observed reflections	575, 575, 544	1028, 1028, 677
R_{int}	0.036	0.104
θ_{max} (°)	31.00	29.57
Refinement on	F^2	F^2
R[F ² >2σ(F ²)], wR(F ²), S	0.034, 0.77, 1.10	0.057, 0.109, 1.16
Data / restraints / parameters	575/ 1/ 37	1028/ 1/ 73
Weighting scheme	$w=1/[\sigma^2(F_o^2)+(0.021P)^2+6.255P]$, where $P=(F_o^2+2F_c^2)/3$	$w=1/[\sigma^2(F_o^2)+(0.036P)^2+25.272P]$, where $P=(F_o^2+2F_c^2)/3$
$\Delta\rho_{\text{max}}, \Delta\rho_{\text{min}}$ (e Å ⁻³)	2.22, -1.66	3.33, -3.08

Table R2. Experimental crystallographic data for NbOCl₂ at 500 K and at room temperature under 5.7 GPa obtained by single-crystal X-ray diffraction.

	NbOCl ₂ , 500K	NbOCl ₂ , 5.7GPa
Chemical formula	NbOCl ₂	NbOCl ₂
M_r	179.81	179.81
Crystal system, space group	Monoclinic, C2/c	Monoclinic, C2/c
a, b, c (Å)	12.9128(15), 3.8993(5), 13.4507(17)	11.699(7), 3.8277(6), 13.157(3)
α, β, γ (°)	90, 105.295(5), 90	90, 106.83(3), 90
V (Å ³)	653.27(14)	563.9(4)
Z	8	8
Density (Mg/m ³)	3.656	4.236
Wavelength (Å)	0.71073	0.71073
μ (mm ⁻¹)	5.044	5.843
Absorption correction	Multi-scan	Multi-scan
T_{\min}, T_{\max}	0.54, 0.85	0.50, 0.88
No. of measured, independent and observed reflections	772, 772, 613	749, 147, 131
R_{int}	0.058	0.042
θ_{max} (°)	27.87	25.30
Refinement on	F^2	F^2
R[F ² >2σ(F ²)], wR(F ²), S	0.078, 0.160, 1.16	0.089, 0.227, 1.25
Data / restraints / parameters	772/ 0/ 37	147/ 24/ 37
Weighting scheme	$w=1/[\sigma^2(F_o^2)+(0.073P)^2+26.668P]$, where $P=(F_o^2+2F_c^2)/3$	$w=1/[\sigma^2(F_o^2)+(0.200P)^2]$, where $P=(F_o^2+2F_c^2)/3$
$\Delta\rho_{\text{max}}, \Delta\rho_{\text{min}}$ (e Å ⁻³)	1.83, -2.22	1.69, -1.24

Table R3. Experimental crystallographic data for NbOI₂ at ambient condition and under 10.7 GPa obtained by single-crystal X-ray diffraction.

	NbOI ₂ , ambient	NbOI ₂ , 10.7 GPa
Chemical formula	NbOI ₂	NbOI ₂
M_r	362.71	362.71
Crystal system, space group	Monoclinic, C2	Monoclinic, C2/m
a, b, c (Å)	15.113(3), 3.9389(8), 7.5284(16)	13.27 (7), 3.7721 (11), 7.222 (6)
α, β, γ (°)	90, 103.845(5), 90	90, 105.6 (3), 90
V (Å ³)	435.13(16)	348.1 (19)
Z	4	4
Density (Mg/m ³)	5.537	6.921
Wavelength (Å)	0.71073	0.56087
μ (mm ⁻¹)	16.751	19.74
Absorption correction	Multi-scan	Multi-scan
T_{\min}, T_{\max}	0.554, 0.600	0.284, 1.000
No. of measured, independent and observed reflections	1971, 797, 677	570, 124, 109
R_{int}	0.036	0.085
θ_{max} (°)	25.7	21.2
Refinement on	F^2	F^2
$R[F^2 > 2\sigma(F^2)], wR(F^2), S$	0.073, 0.151, 1.09	0.094, 0.240, 1.10
Data / restraints / parameters	797/ 7/ 38	124/ 24/ 26
Weighting scheme	$w=1/[\sigma^2(F_o^2)+146.4739P]$, where $P = (F_o^2 + 2F_c^2)/3$	$w=1/[\sigma^2(F_o^2)+(0.2P)^2]$, where $P = (F_o^2 + 2F_c^2)/3$
$\Delta\rho_{\text{max}}, \Delta\rho_{\text{min}}$ (e Å ⁻³)	4.68, -4.54	4.59, -2.75

Table R4. Sample data and structure refinement for NbOI₂ at ambient condition.

Empirical formula	NbOI ₂	
Formula weight	362.71	
Temperature	296(2) K	
Wavelength	0.71073 Å	
Crystal system	Monoclinic	
Space group	C2	
Unit cell dimensions	a = 15.113(3) Å	α = 90°
	b = 3.9389(8) Å	β = 103.845(5) °
	c = 7.5284(16) Å	γ = 90°
Volume	435.13(16) Å ³	
Z	4	
Density (calculated)	5.537 Mg/m ³	
Absorption coefficient	16.751 mm ⁻¹	
F (000)	620	
Crystal size	0.040 x 0.035 x 0.032 mm ³	
θ range for data collection	2.78 to 25.66°	
Index ranges	-18<=h<=18, -4<=k<=4, -9<=l<=9	
Reflections collected	1971	
Independent reflections	797 [R(int) = 0.0358]	
Coverage of independent reflections	98.8 %	
Refinement method	Full-matrix least-squares on F ²	
Data / restraints / parameters	797 / 7 / 38	
Goodness-of-fit	1.094	
Final R indices [$>2\sigma(I)$]	R1 = 0.0727, wR2 = 0.1380	
R indices [all data]	R1 = 0.0899, wR2 = 0.1512	
Largest diff. peak and hole	4.680 and -4.540 e·Å ⁻³	

$R = \Sigma||F_o| - |F_c|| / \Sigma|F_o|$, $wR = \{\Sigma[w(|F_o|^2 - |F_c|^2)^2] / \Sigma[w(|F_o|^4)]\}^{1/2}$ and calc
 $w = 1/[\sigma^2(F_o^2) + 146.4739P]$ where $P = (F_o^2 + 2F_c^2)/3$

Table R5. Atomic coordinates and equivalent isotropic atomic displacement parameters (Å²) for NbOI₂ at ambient condition.

Label	x	y	z	Occupancy	Site	U _{eq} *
I001	0.6499 (2)	0.2350 (10)	0.0713 (5)	1	4c	0.0282 (8)
I002	0.6335 (2)	0.2326 (9)	0.5605 (5)	1	4c	0.0286 (8)
Nb03	0.5005 (3)	0.2852 (8)	0.2215 (9)	1	4c	0.0384 (14)
O004	0.500 (2)	0.751 (11)	0.221 (4)	1	4c	0.037 (8)

*U_{eq} is defined as one third of the trace of the orthogonalized U_{ij} tensor.

Table R6. Anisotropic atomic displacement parameters (\AA^2) for NbOI₂ at ambient condition.

Label	U ₁₁	U ₂₂	U ₃₃	U ₁₂	U ₁₃	U ₂₃
I001	0.0483 (18)	0.0131 (14)	0.0274 (16)	0.001 (2)	0.0175 (15)	-0.001 (2)
I002	0.0534 (19)	0.0135 (15)	0.0176 (14)	0.003 (3)	0.0057 (15)	-0.001 (2)
Nb03	0.0193 (18)	0.007 (2)	0.086 (4)	0.003 (2)	0.007 (3)	0.004 (3)
O004	0.027 (14)	0.023 (14)	0.06 (2)	0.01 (2)	0.002 (16)	0.01 (2)

The anisotropic displacement factor exponent takes the form: $-2\pi^2[h^2a^{*2}U_{11} + \dots + 2hka^*b^*U_{12}]$.

Table R7. Bond lengths (\AA) for NbOI₂ at ambient condition.

Label	Distances	Label	Distances
I001—Nb03	2.764 (6)	Nb03—O004	1.83 (5)
I001—Nb03 ⁱ	2.769 (7)	Nb03—O004 ⁱⁱⁱ	2.11 (5)
I002—Nb03	2.854 (7)	Nb03—Nb03 ⁱ	3.331 (14)
I002—Nb03 ⁱⁱ	2.903 (7)		

Symmetry codes: (i) $-x+1, y, -z$; (ii) $-x+1, y, -z+1$; (iii) $x, y-1, z$; (iv) $x, y+1, z$.

Table R8. Bond angles ($^\circ$) for NbOI₂ at ambient condition.

Label	Angles	Label	Angles
Nb03—I001—Nb03 ⁱ	74.0 (2)	O004—Nb03—I002 ⁱⁱ	93.8 (11)
Nb03—I002—Nb03 ⁱⁱ	93.6 (2)	O004 ⁱⁱⁱ —Nb03—I002 ⁱⁱ	85.7 (9)
O004—Nb03—O004 ⁱⁱⁱ	179.2 (19)	I001—Nb03—I002 ⁱⁱ	167.1 (2)
O004—Nb03—I001	94.4 (11)	I001 ⁱ —Nb03—I002 ⁱⁱ	83.93 (16)
O004 ⁱⁱⁱ —Nb03—I001	86.2 (9)	I002—Nb03—I002 ⁱⁱ	85.8 (2)
O004—Nb03—I001 ⁱ	93.7 (10)	O004—Nb03—Nb03 ⁱ	90.0 (11)
O004 ⁱⁱⁱ —Nb03—I001 ⁱ	85.6 (9)	O004 ⁱⁱⁱ —Nb03—Nb03 ⁱ	90.0 (9)
I001—Nb03—I001 ⁱ	105.4 (2)	I001—Nb03—Nb03 ⁱ	53.05 (17)
O004—Nb03—I002	94.5 (10)	I001 ⁱ —Nb03—Nb03 ⁱ	52.93 (19)
O004 ⁱⁱⁱ —Nb03—I002	86.1 (9)	I002—Nb03—Nb03 ⁱ	136.8 (3)
I001—Nb03—I002	83.73 (16)	I002 ⁱⁱ —Nb03—Nb03 ⁱ	136.9 (3)
I001 ⁱ —Nb03—I002	167.2 (2)	Nb03—O004—Nb03 ^{iv}	179.2 (19)

Symmetry codes: (i) $-x+1, y, -z$; (ii) $-x+1, y, -z+1$; (iii) $x, y-1, z$; (iv) $x, y+1, z$.

REVIEWERS' COMMENTS

Reviewer #2 (Remarks to the Author):

The authors have addressed my previous comments in the reasonable and acceptable forms. No further revisions are needed.

Response Letter

We thank the referee for his/her careful and professional review on the revised manuscript. We have carefully considered the comment and made a response as follows.

Response for the referee:

Comment 1: “The authors have addressed my previous comments in the reasonable and acceptable forms. No further revisions are needed.”

Response 1: We thank the referee again for recommending this manuscript to be published on *Nat. Commun.*